# Evaluating the quality of remote sensing products for agricultural index insurance

**Benson K. Kenduiywo**[1,2]*, **Michael R. Carter**[3], **Aniruddha Ghosh**[1,4], **Robert J. Hijmans**[1]

**1** Department of Environmental Science and Policy, University of California, Davis, CA, United States of America, **2** Department of Geomatic Engineering and Geospatial Information Systems, Jomo Kenyatta University of Agriculture and Technology, Nairobi, Kenya, **3** Department of Agricultural and Resource Economics, University of California, Davis, CA, United States of America, **4** International Center for Tropical Agriculture, Nairobi, Kenya

* bkenduiywo@jkuat.ac.ke

## Abstract

Agricultural index insurance contracts increasingly use remote sensing data to estimate losses and determine indemnity payouts. Index insurance contracts inevitably make errors, failing to detect losses that occur and issuing payments when no losses occur. The quality of these contracts and the indices on which they are based, need to be evaluated to assess their fitness as insurance, and to provide a guide to choosing the index that best protects the insured. In the remote sensing literature, indices are often evaluated with generic model evaluation statistics such as $R^2$ or Root Mean Square Error that do not directly consider the effect of errors on the quality of the insurance contract. Economic analysis suggests using measures that capture the impact of insurance on the expected economic well-being of the insured. To bridge the gap between the remote sensing and economic perspectives, we adopt a standard economic measure of expected well-being and transform it into a Relative Insurance Benefit (*RIB*) metric. *RIB* expresses the welfare benefits derived from an index insurance contract relative to a hypothetical contract that perfectly measures losses. *RIB* takes on its maximal value of one when the index contract offers the same economic benefits as the perfect contract. When it achieves none of the benefits of insurance it takes on a value of zero, and becomes negative if the contract leaves the insured worse off than having no insurance. Part of our contribution is to decompose this economic well-being measure into an asymmetric loss function. We also argue that the expected well-being measure we use has advantages over other economic measures for the normative purpose of insurance quality ascertainment. Finally, we illustrate the use of the *RIB* measure with a case study of potential livestock insurance contracts in Northern Kenya. We compared 24 indices that were made with 4 different statistical models and 3 remote sensing data sources. *RIB* for these indices ranged from 0.09 to 0.5, and $R^2$ ranged from 0.2 to 0.51. While *RIB* and $R^2$ were correlated, the model with the highest *RIB* did not have the highest $R^2$. Our findings suggest that, when designing and evaluating an index insurance program, it is useful to separately consider the quality of a remote sensing-based index with a metric like the *RIB* instead of a generic goodness-of-fit metric.

**Data Availability Statement:** The data underlying the results presented in the study are available from Github (https://github.com/reagro/agrodata). Tutorials on how to use the data and the tools we

developed are also available on this link https://reagro.org/ as noted on the Github data page.

**Funding:** This work has been supported by the "Innovations to Improve the Quality and Uptake of Agricultural Index Insurance" grant provided to the BASIS Markets, Risk and Resilience Innovation Lab funded by USAID under grant no. 720BFS18CA00001.

**Competing interests:** The authors have declared that no competing interests exist.

## Introduction

Agricultural index insurance can reduce the negative consequences of adverse weather such as droughts in crop and livestock production. Apart from the direct benefits of alleviating the risk of a sudden loss of income or capital, insurance also encourages producers to invest more in agriculture, and increase their welfare in normal years as well while alleviating poverty traps [1–5]. Given that insurance is not free, poorly designed programs are not useful, or even harmful as they can leave farmers worse off than without insurance if they do not receive payouts when they should and need them most [6]. Payouts are triggered when a pre-defined index is below (or above) a certain threshold, and having a reliable index is thus essential for a successful index insurance product. See [6] for other aspects of the design of an index insurance program that we do not discuss here.

Indices can be based on direct observations of productivity, such as estimating crop yield through crop-cuts, or on a proxy associated with productivity such as rainfall at a weather station, or a measure derived from remote sensing data. Because of its low cost and large coverage, remote sensing derived indices are increasingly used [7, 8]. It is, however, important to evaluate the quality of remote sensing-based indices to assess if they perform well enough for the purposes of insurance programs and to choose between different candidate indices.

A remote sensing-based index that estimates, for example, crop yield, can be evaluated with generic measures of goodness of fit such as the coefficient of determination ($R^2$) or the Root Mean Square Error (RMSE) that compare observed with predicted yield. However, except for the extreme cases of almost no fit or near perfect fit, these measures are not directly informative about their value to the insurance program. While a higher score for a relevant statistic might always seem preferable, a particular value does not directly indicate whether a remote sensing-based index is fit for purpose, nor does it guarantee that the model with the highest goodness of fit is the most useful. For example, two models might have the same goodness of fit, but if one is better at predicting crop yields when they are low (which is relevant for insurance) and the other can more accurately predict high crop yields (which is not relevant), the former model would be more useful for insurance purpose. One approach to make the evaluation of indices more relevant would be to use an asymmetric goodness of fit measure that weighs certain errors (when low yield is observed) more than other errors (when high yield is observed). There are several ways to do this but the connection between the goodness of fit measure and the quality of the insurance program would still be unclear. For that reason, we developed a metric that is more informative to evaluate remote sensing indices as it directly assesses their value to the insurance product relative to a perfect index that has no error. In doing so, we draw on economic approaches that suggest alternative ways of measuring insurance value (*e.g.*, [9]).

While the economics literature offers a number of metrics that can be used to measure index insurance quality, *RIB* relies on expected utility theory as its underlying normative framework to judge quality. After introducing the risk and insurance problem that confronts the pastoralist population that we use to illustrate the use of *RIB*, we present the expected utility framework underlying *RIB*, discuss its desirable properties as a normative measure, and show that it can be decomposed into comprehensible set of common-sense indicators that help us understand whether, when and why a particular insurance contract is of high or low quality. We then compute *RIB* values for a number of indices in a case study on a hypothetical livestock insurance program in northern Kenya, contrasting *RIB* values with predictive skill and model evaluation statistics more commonly used in the remote sensing literature, exploring how and why the two results obtained with the two metrics differ. We conclude with observations on the use of the *RIB* measure to certify quality and aid in the design of an index insurance that

best promotes the economic well-being of the population at risk that the insurance is designed to protect. Below we first describe this economic framework to show how the *RIB* is derived.

## Evaluation framework

This section introduces basic insurance and economic concepts that can be used to evaluate the insurance benefit offered by remote sensing-based insurance contracts based on [6]. The section uses a series of graphical devices in order to explain the concepts and shows how we derived our new measure.

**Risk and the insurance problem in the pastoralist regions of Kenya.** To illustrate the framework, we first consider a perfect livestock asset insurance contract for pastoralist households in our study region in Northern Kenya. These households economically depend almost exclusively on their livestock, which constitute more than 90% of total household wealth and generate 80% of household income. Livestock wealth is typically measured in Tropical Livestock Units (TLU), a measure that allows counts of different livestock species to be combined into a single number. In this convention, one TLU is equivalent to 1 cow, 0.7 camels, or 10 goats or sheep. While all but the most indigent households possess multiple TLU, to keep things simple here we analyze the case of a household that has only 1 TLU. In monetary terms, we will assume that 1 TLU is worth $US 1000, an amount roughly in line with animal prices during non-drought periods.

Pastoralism is risky in arid and semi-arid climates like in our study region. Periodic droughts lead to livestock starvation and death, sometimes destroying more than half of a family's livestock over the course of a few months. These large decreases in productive wealth generate large drops in income and increases in human suffering as families struggle to eat. For our illustrative household (denoted with the subscript $i$) that begins with 1 TLU of livestock wealth or capital, and is not insured (denoted with the superscript $N$) the amount of wealth they have the next season ($k_{i2}^N$) depends, in the absence of insurance, on the livestock mortality they experience ($M_i$). We can thus write next season's livestock held by the household $i$ as:

$$k_{i2}^N = k_{i1}(1 - M_i) = (1 - M_i), \text{for household with } 1\ TLU \tag{1}$$

where $k_{i1}$ is the household's initial livestock, the mortality rate $M_i$ ranges between 0 (no mortality) and 1 (all animals die). Because we assume that $k_{i1} = 1$ for our example, the expression can be further simplified as shown in Eq (1). Note that to keep simplify our discussion we are ignoring the natural reproductive growth of the herd.

Using the data described in the methods section below, we estimate the probability distribution of livestock mortality and use it to calculate the probability that our stylized pastoralist household will enter next season with the different amounts of livestock wealth shown in Fig 1. The histogram projected on the figure reflect the different probabilities the household faces. For example, the right most bar represents a near zero mortality rate, which occurs just under 5% of the time. In this case, the household would enter the next season with its $1000 in productive assets intact. At the other extreme, the leftmost bar represents a 65% mortality rate. While such extreme events happen less than 1% of the time, when they do occur the household would enter the next season with only $350 of livestock wealth as shown in the figure. More generally, Fig 1 displays the probability that the stylized household will enter the next season with the different amounts of livestock wealth shown on the horizontal axis. As can be seen, the risk of loss is substantial, suggesting that insurance could play an important role for pastoralists households.

**Payoff functions and the pricing of insurance.** A well-functioning contract indemnifies households for losses and thereby puts a floor under household asset holdings. Fig 1 illustrates

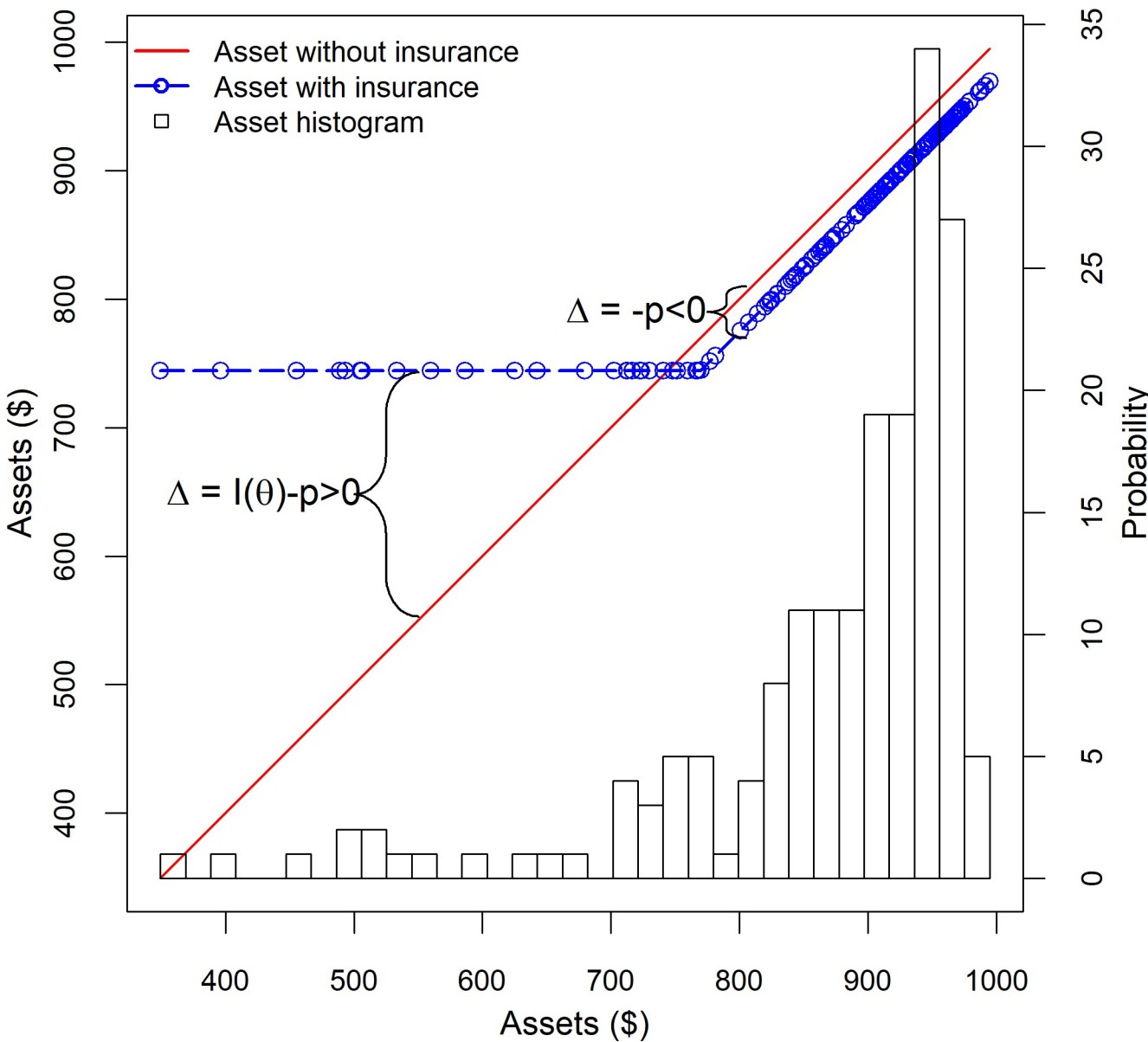

**Fig 1. Histogram showing livestock mortality in different asset bins, household livestock assets without insurance, and assets with a perfect insurance contract.** Each observation is a random draw from an underlying risk distribution for a household with about 1 tropical livestock unit expressed in US dollars ($); in this simplified example we only consider mortality, not herd growth through birth. See text for a description of the equations.

how a "perfect" insurance contract would work. By a perfect insurance contract, we mean a contract that observes and issues payouts based on the actual livestock mortality, $M_i$, that a household experiences (later on we consider more realistic contracts where $M_i$ has to be estimated because it is prohibitively expensive to measure mortality for each and every household). We assume that this contract carries a 23% deductible, but then issues payments to compensate the household for any losses in excess of 23% according to the following perfect insurance payoff or indemnity function:

$$I_i^P(M_i) = \max\{0, M_i - t\} \times k_{i1}, \qquad (2)$$

where the superscript *P* stands for perfect insurance contract and *t* is the deductible (sometimes called the payout trigger) which we set to 23% in our example. $k_{i1}$ is the number of livestock units insured, which we assume to be 1 TLU in our example. Under this payoff function, the insurance pays nothing if the mortality rate is less than the trigger *t*, that is, $t<23\%$ in our example, and otherwise pays enough to restore the value of all livestock back to 77% of its initial level. As can be seen in Fig 1, insurance puts a floor under the household asset holdings such that they will never fall below $744 in the subsequent season.

To pay for this protection, every season the household pays a premium, *p*. The starting point for calculation of an insurance premium is the "actuarially fair price" (AFP), which is defined as the long-run average or expected indemnity payout per-TLU insured. In our example, we estimate that the mortality rate can take on 25 different values as shown in Fig 1. We denote each of these values as $M_{ij}$ and we denote $\pi_j$ as the probability that mortality rate *j* occurs. Using this notation, we can write the AFP for our perfect insurance contract as:

$$\text{AFP}^{\text{P}} = \sum_{j=1}^{25} I^P(M_{ij})\pi_j. \tag{3}$$

Unless subsidized, insurance is always sold at some mark-up rate, *m*, that covers the operating costs of the insurance provider. The final market price faced by the household can be expressed as follows:

$$p^P = \text{AFP}^{\text{P}}(1 + m). \tag{4}$$

In our case, given the estimated probability distribution, $AFP^P$ = $20 and we assume it is marked up by a further 25%. When the family purchases insurance it reduces its wealth by $25. The payment of this premium is visible in Fig 1 as the reduction in wealth that the household would face in years of low mortality when no indemnity payments are received. When the stylized household purchases insurance, its livestock wealth next season will be:

$$k_{i2}^P(M_i) = (1 - M_j) + I^P(M_j) - p^P. \tag{5}$$

Insurance provides economic benefit to households by putting a floor or safety net by transferring resources from good times (when premiums are paid and no payouts received), when resources are relatively plentiful, to bad times, when resources are scarce and especially valuable to the household. The value of perfect insurance is intuitive from the wealth floor in Fig 1. The household sacrifices wealth in the good years in order to avoid the draconian consequences of severe losses in bad years. We define the difference between the household's wealth with and without insurance as Δ.

**Valuing the quality of an insurance contract.**   While the value of insurance to the household is fairly intuitive in the case of the perfect insurance contract illustrated in Fig 1, we will later consider imperfect contracts, which sometimes fail to pay when the household has losses (*false negatives)*, and sometimes pays when households do not (*false positives)*. In these cases, it is more difficult to discern when a contract is good enough to be worth buying [1], and even harder to judge which of several alternative contracts offers better economic support for households. In other words, we need a cogent standard to measure the quality of protection offered by an insurance contract, and to evaluate and select the remote sensing-based indices that it uses.

A number of studies in the economics literature have employed different metrics which speak to the quality of index insurance contracts. These can be grouped into (i) Measures that examine the impact of the insurance on some feature of the probability distribution for wealth;

and, (ii) Measures that are based on an explicit normative or welfare metric designed to capture the economic well-being of the insured household. Studies in the first category include [10], who study the hedging effectiveness of insurance, and a number of studies that look at the risk-reducing potential of insurance [11–14]. In a similar spirit, [15] study the catastrophic performance ratio (defined as expected payouts, normalized by the sum insured, in catastrophic, left tail states of the world). Somewhat similarly, [9] study the impact of insurance on lower partial moments of the probability distribution. While these approaches all offer valuable insights, they focus on changes in the left tail portion of the probability distribution. While what happens in the left tail is very important for the value of insurance, index contracts that incorrectly issue payouts in the right tail (false positives) are also damaging to the welfare value of insurance. This because as we discuss more below, it always costs more than $1 to get $1 of a payout. Paying more than a dollar to get a dollar makes sense if a dollar is worth more than a dollar, as it is in bad states of the world. In good states of the world, however, a dollar is worth only a dollar and paying, say, $1.25 to get a $1 does not improve economic welfare.

The second type of quality measures—those based on explicit measures of individual well-being—consider the full distribution of outcomes and avoid the incompleteness of lower tail measures. As measures of well-being, they also open the door to a natural quality measure: a good insurance contract should increase the individual's level of expected well-being compared to her well-being without insurance ([6] call this a minimum quality standard). While there are several well-developed economic approaches to measuring individual well-being in the face of risk, we will first explain the use of "expected utility" normative framework to measure quality and then discuss its strengths relative to alternative approaches.

The economist's standard utility function expresses economic well-being, or utility, $U$ as a function of available purchasing power or wealth. It is conventionally assumed that people always prefer to have more wealth or purchasing power. That is, additional wealth makes a household economically better off, and the additional value to the household of additional wealth (or "marginal utility") is strictly positive (mathematically, that $^dU/_dk > 0$). We label this marginal value of additional wealth as the shadow value of wealth or money as it represents the additional level of economic well-being a household can achieve with one more dollar of wealth. We denote this shadow value of purchasing power or money as $\lambda(k)$, noting the value of additional wealth depends on how much wealth the household has.

The conventional assumption in economics is that as we become richer, the additional value of another dollar of wealth stays positive, but becomes smaller (i.e., marginal utility diminishes, or mathematically that $\frac{d^2U}{dk^2} < 0$). In other words, $\lambda(k)$ becomes smaller as we get richer. Flipped around, this same shadow value sensibly increases as wealth decreases and the family becomes more desperate (a dollar is worth more in times of economic stress than in better times). It is the curvature of the utility function which ultimately creates the asymmetric loss function that underlies the value of insurance measures defined below.

In the analysis that follows, we assume that the utility function follows a specific functional form (known as a constant relative risk aversion utility function) that conforms to the assumptions just discussed:

$$U(k) = \frac{k^{(1-\rho)}}{(1-\rho)}, \; where \; \rho \neq 1 \tag{6}$$

where the parameter $\rho$ shapes the degree of sensitivity to risk and bad outcomes. Fig 2 shows how $\lambda$ changes as wealth changes for the case where $\rho = 2$. As $\rho$ increases, $\lambda$ increases more steeply as wealth diminishes. Throughout the analysis here, we assume that $\rho = 2$, an

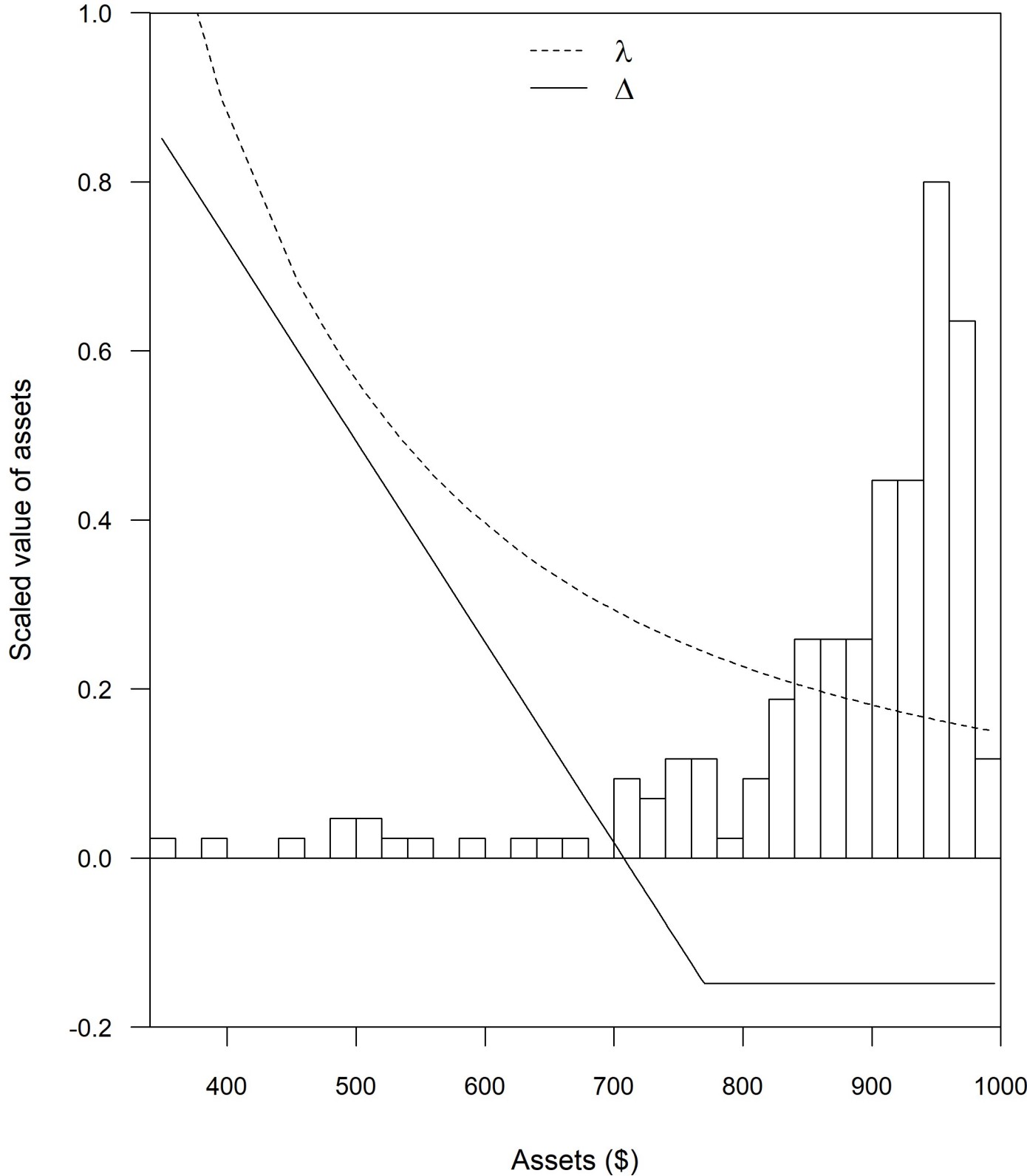

**Fig 2. Elements of insurance benefit: The shadow value of money ($\lambda$), insurance benefit ($\Delta$), and an empirical distribution (histogram) of the probability of the livestock assets remaining after accounting for mortality expressed in USD.** To combine these measures on one figure, the y-axis was scaled to values shown on the figure.

assumption supported by experiments that measure $\rho$ for small scale farmers in developing countries (see [16]).

While the utility function given in Eq (6) gives us the economic well-being associated with a given outcome, when considering risk, we need to consider a multiplicity of possible outcomes that might occur. When considering the economic value of a risky prospect (*e.g.*, well-being in a world without insurance), the conventional economic analysis employed by [6] suggests valuing that prospect by the average or expected utility that might occur given the underlying random variable (livestock mortality in our case). Average or expected utility is the sum of all the possible utility levels that might occur, each weighted by their probability of occurrence. Formally, expected utility (or expected economic well-being) for our stylized household with 1 TLU, but no insurance, $EU^N$, is calculated as follows:

$$EU^N = \sum_{j=1}^{25} U(1(1 - M_j))\pi_j, \tag{7}$$

while expected utility for the case of perfect insurance can be written as:

$$EU^P = \sum_{j=1}^{25} U(1(1 - M_j) + I^P(M_j) - p^P)\pi_j. \tag{8}$$

Using these expressions, we can define the insurance benefit to the household under perfect insurance ($IB^P$) as the difference between their expected well-being with and without perfect insurance:

$$IB^P \equiv EU^P - EU^N \tag{9}$$

Analogue expressions to (8) and (9) can be defined for any (imperfect) index insurance contract J. A minimum quality standard for that alternative imperfect insurance contract would be that $IB^J > 0$, *i.e.*, insurance does not harm and at least improves expected well-being of the insured household [6]. Although $IB^J > 0$ is clearly a minimum standard (ideally the insurance benefits should not only be positive, but also be sufficiently large to be meaningful), it is a non-trivial one to fulfill as illustrated by the failure of the rainfall-based index insurance contract discussed in [1].

While expected utility theory has deep roots within the discipline of economics, its *descriptive* accuracy has been called into question by a range of experimental studies that show that some people do not make choices that conform with those that would maximize their expected utility. In efforts to accommodate systematic behavioral deviations from the predictions of expected utility theory [17] and [18] assembled alternative theories of how individuals make decisions in the face of risk, known as cumulative prospect theory (CPT) and rank dependent utility (RDU). These two alternative frameworks share the perspective that people make decisions using a probability weight that may differ systematically from objective probabilities. In addition, CPT assumes that individuals value losses differently than gains, creating a kink in the smooth utility function shown in Eq (6), where the kink occurs at what they call the reference point that distinguishes losses from gains. While such a reference point can be easily established in a behavioral experiment, its definition is less obvious in real world circumstances, making it more difficult to use CPT as a general tool for evaluating the quality of index insurance [19]. While CPT and RDU approaches have both been used to evaluate the value of index insurance to an individual (see [20, 21]), it is not obvious that a welfare metric based on misperception of probabilities is appropriate to judge and design insurance quality. Put differently, while these alternative approaches may be descriptively more accurate than expected

utility theory in predicting insurance uptake, it is not apparent that they form the better basis for normative judgements [17, 19] especially if the source of objectively wrong probability weights is simple misperception.

While these issues of normative bases for value judgements are philosophically complex, in the remainder of this paper we will maintain our focus on expected utilty based quality metrics as those metrics seem most defensible for the purposes at hand. In addition, while all this discussion may seem obtuse to non-economists, the expected utility based quality metric in Eq (9) above has the additional virtue that it can be decomposed into three parts that most economist would agree should underwrite a conceptually sound measure of insurance quality:

1. $\Delta(M_j)$, the difference in final wealth with and without insurance under each possible mortality rate, where $\Delta(M_j) = k^P(M_j) = k^N(M_j) = I^P(M_j) - p^P$ (Fig 1),

2. $\lambda(\theta_j)$, the shadow value of money when the difference occurs; and,

3. $\pi_j$, the probability that a given mortality rate and difference in wealth occurs.

Specifically, using a first order Taylor Theorem approximation of Eq (9), the quality metric for the perfect insurance contract, $IB^P$, can be approximated as:

$$IB^P \approx \sum_{j=1}^{25} \Delta(M_j) \times \lambda(M_j) \times \pi_j. \tag{10}$$

In words, the quality metric, $IB^P$, is just the average (probability-weighed) difference in wealth with and without insurance times the shadow value of money when the difference occurs. Fig 2 graphs the elements of this expression. Note that $\Delta(M_j)$ will most frequently be negative because in most seasons an insured household is expected to pay a premium but not to receive a payout, and this will drive $IB$ downwards. However, when insurance works well, the shadow value of money ($\lambda(M_j)$) will be low when $\Delta$ is negative, but $\lambda$ will be high when $\Delta$ is positive. As shown by the graph, the value of insurance can only be positive if insurance correctly pays off when things are bad and $\lambda$ is high. As we will see below with actual examples of imperfect insurance, the economic value of insurance is reduced substantially if the insurance fails to pay off during high mortality shocks when $\lambda$ is high. Indeed $IB$ can be negative if the contract is of a poor enough quality, meaning the household would be better off not buying the insurance.

While the basic quality metric, $IB^J$, can be used to judge whether any particular contract $J$ is good enough to provide economic benefit to the insured household, we propose a relative quality measure that will allow choosing between different alternative remote sensing indices (based on sensor(s) employed, statistical model used to predict losses based on sensor readings, etc.). Before forming that measure, we first transform the underlying expected utility measures, $EU^N$ and the $EU^J$ into more interpretable "certainty equivalent income" metrics. In words, the certainty equivalent of the pastoralist's expected utility without insurance, denoted $CE^N$ is the amount of income that if received for certain every year, would give the same pastoralist expected utility level $EU^N$. The certainty equivalent income for the no insurance case is calculated as

$$CE^N = [(1 - \rho)EU^N]^{\frac{1}{1-\rho}}$$

Analogue expressions can be used to calculate the certainty equivalent income for the case when the pastoralist has insurance. Using our data, $CE^N = \$848$ for a moderately risk averse person who has a coefficient of relative risk version of 2. This amount is below the expected value of wealth (\$863), indicating that the person would take a guaranteed wealth below their

expected wealth to avoid the severe suffering that attends years in which the livestock mortality rate is high.

After calculating the same object for the perfect insurance case, we can define a transformed analogue to Eq (9) which we write as:

$$\widehat{IB}^P = CE^P - CE^N. \tag{11}$$

Using these transformed measures, we now propose a Relative Insurance Benefit (*RIB*) measure that allows us to benchmark any feasible insurance contract *J* against the insurance benefit offered by the perfect, failure proof, insurance contract:

$$RIB^J = \frac{\widehat{IB}^J}{\widehat{IB}^P}, \ where \ \widehat{IB}^P > 0, \tag{12}$$

where $RIB^J$ is the relative insurance benefit of contract *J*. The measure is only meaningful in the case where perfect insurance offers benefits to insured households ($IB^P > 0$). Note that $RIB^J \leq 1$, with a value of 1 indicating the contract is just as good as perfect insurance and a value less than zero indicating that the contract *J* does not even pass the minimum quality standard test and offers no value to the household. An alternative contract design *K*, which has $RIB^K > RIB^J$, would be strictly preferred to contract *J* based on this insurance quality metric. More generally, the remote sensing challenge is to find an index that pushes *RIB* as close to one as possible. In the analysis to follow, we will show that while related to predictive skill measures that are often used to evaluate alternative contract designs, the conceptually-based *RIB* measure captures relevant features of contract design that predictive skill measures do not.

## Materials and methods

In the prior section, we focused on a hypothetical perfect insurance contract in which we assumed the livestock losses of each household were directly observed and insurance payments based on those observations. For low wealth and isolated households, such direct measurement is not economically feasible and hence the interest in relying on remote sensing measures that can estimate livestock losses, with these estimates serving as the basis for indemnity payments [7, 22]. To design a remote sensing index and measure its quality requires ground data on actual livestock losses as well as remote sensing data that can be used to predict those losses.

### Study area

We chose Marsabit, a county in northern Kenya, as the study region because of availability of household survey data on livestock loss from long term operational livestock insurance programs in this region [12, 23, 24]. Nomadic pastoralism is the dominant agricultural activity and the main source of income for most communities [25]. The climate in Marsabit is arid to semi-arid, with an average annual rainfall of 400–800 mm [26]. The rainfall pattern is bimodal, with a slightly longer wet period between March and May and a shorter period between October and December. From a herding perspective that makes for two annual seasons, the 7-months Long Rain–Long Dry season (LRLD) from March to September and the 5-months Short Rain–Short Dry season (SRSD) from October to February. This study uses data aggregated by Marsabit's former administrative district boundaries that included 24 sub-locations. To determine when insurance payouts should occur, and how much should be paid, we modelled livestock loss as a function of the remote sensing indices (rainfall and Normalized Difference Vegetation Index (NDVI), non-transformed and log-transformed).

## Livestock mortality data

We used livestock mortality data collected from households in 15 sub-locations of Marsabit by the International Livestock Research Institute [27]. Annual household surveys in Marsabit were conducted between October and November to cover the years 2008–2015. Each of the annual surveys was carried out after IBLI insurance sales in each corresponding season (i.e. LRLD or SRSD). The main purpose of the data collection was to help develop and monitor the Index Based Livestock Insurance (IBLI) program in Marsabit [27]. The number of household surveyed per year varied between 1739 and 3549. Surveys were carried out between October and November and included questions related to household demographic information, livestock production, livelihoods activities and income sources, expenditures and consumption, health and educational outcomes, assets, access to credit, market interaction, and community poverty rates. These data have been described in detail by [27] and also used by [5, 12, 13, 23, 28–31] to design and evaluate impact of IBLI program.

Of prime interest to our study was the livestock loss data. Livestock loss can occur for several reasons, but loss from drought-related starvation was the primary reason, followed, at some distance by disease. To combine data on different livestock species, the livestock numbers were transformed to the equivalent TLUs. We averaged mortality data by a given sub-location and season. Overall the average morality in rate from the survey data was 13%.

## Remote sensing data

We used rainfall and NDVI data to create indices. For rainfall, data from CHIRPS [32] with a 0.05° (approximately 5.5 km at the equator) spatial resolution was used. We had two different sources of NDVI data: i) daily NOAA Advanced Very High Resolution Radiometer (AVHRR) NDVI records, at a spatial resolution of 0.05° for the period between 1982 and 2019 [33] and ii) NDVI computed from red and near infra-red bands of 8-day 500 m spatial resolution MOD09A1 surface spectral reflectance product for 2001 to 2019. We masked out clouds, shadows, and surface water in both NDVI datasets.

The NDVI data from MODIS (MD) and NOAA (NO) and the rainfall (RN) were spatio-temporally aggregated by computing the mean value for each sub-location and for each season (LRLD and SRSD). We log-transformed the data to make them more normally distributed and refer to these as LMD (MODIS), LNO (NOAA) and LRN (rainfall). To account for variation between sub-locations in the mean rainfall and NDVI scores we computed $z$-scores (Eq (13)) for the non-transformed and for the log-transformed aggregated data [34]. $z$-scores enables the quantification of a particular index's deviations from a long term mean that is independent of the size of that mean. $z$-scores were computed as:

$$z_{\ell s} = \frac{index_{\ell s} - \mu_\ell}{\sigma_\ell} \tag{13}$$

where $index_{\ell s}$ is the mean value of a particular index for a sub-location $\ell$ in a season $s$, while $\mu_\ell$ and $\sigma_\ell$ represents the mean and standard deviation across all years for the corresponding location.

## Livestock loss models

To illustrate the use of the *RIB* measure, we use the data on each candidate remote sensing index to estimate a set of 4 regression models that estimate livestock mortality in location $\ell$ in season $s$, $M_{\ell s}$. Note that the procedures we are using parallel the actual approach that was used to create the original IBLI contract in northern Kenya, although that original exercise was based on

different data sources and a more restricted set of remote sensing based measures [23]. The regression models are standard linear regression (lm), piecewise linear regression when $z$-scores are less than zero (lm0); piecewise linear regressions when $z$-scores are less than -0.5 (lm5); and, a segmented regression (sm) that estimates the optimal breakpoint. Formally we estimate:

$$lm : \; M_{\ell s} = \beta_1 z_{\ell s} + \beta_0 + \varepsilon_{\ell s}, \tag{14}$$

$$lm0 : \; M_{\ell s} = \beta_1 z_{\ell s} + \beta_0 + \varepsilon_{\ell_s} \; estimated \; for \; z_{\ell s} < 0, \tag{15}$$

$$lm5 : \; M_{\ell s} = \beta_1 z_{\ell s} + \beta_0 + \varepsilon_{\ell s} \; estimated \; for \; z_{\ell s} < -0.5, \tag{16}$$

and

$$sm : \; M_{\ell s} = \beta_1 z_{\ell s} + \beta_{02}(z_{\ell s} - \psi)I(z_{\ell s} > \psi) + \beta_0 + \varepsilon_{\ell s}. \tag{17}$$

For the segmented model [35], the parameter $\psi$ represents the breakpoint where the slope of the mortality regression changes and $I(z_{\ell s} > \psi)$ is an indicator function that takes on the value for those observations for which the inequality condition holds. This segmented model is very similar to the model used for the initial IBLI contract in northern Kenya [23]. While that earlier work was not guided by the sort of explicit quality standard we propose here, we shall see that this model is in fact our best performing model using the *RIB* standard. In total, we estimate 24 models based on the six remote sensing measures and the 4 regression methods.

Several recent contributions to the literature [9, 36] recommend conditional quantile methods to estimate loss functions such as Eq (14). Conditional quantile analysis captures the regression function parameters that best characterize the data in any percentile of the conditional error distribution of $\varepsilon_{it}$. Following this recommendation, we would estimate the parameters in 77[th] percentile of that conditional error distribution, thus capturing the impact of the forage index on mortality losses for those observations that had larger losses than what would expect given the level of the index. While it could be that these large conditional loss observations are characterized by excess sensitivity to rangeland conditions, it may also be that these large conditional losses are explained by factors other than forage scarcity (e.g., disease). Hence, it is not apparent that the sensitivity of these observations to the index is necessarily informative about the sensitivity of mortality rates to the forage index. We do incorporate these quantile methods into our analysis but discuss our specific findings regarding the quality of quantile-based estimation.

## Payout function and measurement of *RIB*

Section 2 details the quality evaluation method that guides our work. While that section largely focused on a perfect insurance contract based on actual livestock losses, it is easily adapted to analyze the candidate index insurance contracts by simply replacing actual livestock mortality with mortality predicted by the candidate model, $\widehat{M}_{\ell s}^{J}$, where $J$ is the indicator of the model. The insurance payout function thus becomes:

$$I_{\ell s}^{J} = \max(0, \; \widehat{M}_{\ell s}^{J} - t) \times k_0, \tag{18}$$

where, as before, $t$ is the trigger or deductible level (set at 23%) and we assume initial livestock ($k_0$) is 1 TLU worth \$1000. Using this indemnity function, we can calculate the actuarially fair insurance price for candidate contract $J$ by averaging the payouts across the years for which we have data available. After applying a 25% markup to that price, Eq (5) allows us to calculate the wealth of the household when it purchases one unit of insurance coverage under contract J.

For each location and season combination for which we have ground data, we can calculate the remaining wealth of the prototypical household after experiencing that season's shock. For that same location and season, we can recover predicted mortality under the candidate contract and calculate the remaining of the household wealth had it purchased insurance contract *J*. Repeating this procedure generates two series of wealth outcomes (one without and one with insurance). Taking the utilities of each wealth value using Eq (6) allows us to calculate the expected utilities of each series by taking the simple average of each the utilities across locations and seasons. The insurance benefit, $IB^J$, of contract *J* can thus be calculated per expression (9). Calculation of the relative insurance benefit measure, $RIB^J$, follows immediately using the relevant expressions presented at the end of section 2 above. These computations were implemented in R package 'agro' [37] that implements these procedures and may be adapted to other data sets.

## Results

This section reviews the performance of each of the 24 candidate index insurance contracts that could be offered to pastoralist households in Northern Kenya. We first examine the basic predictive skill of each the 24 models, summarizing its predictive performance using a conventional $\boldsymbol{R}^2$ goodness of fit measure. We then contrast the models' predictive performance with its quality as measured by *RIB*. While the measures are correlated, we look in detail at several cases where the rankings by predictive skill and *RIB* are quite different. These comparisons allow us to see how and why the measures are different and help substantiate the value addition of *RIB* as a tool to design and evaluate index insurance contracts.

### Estimating and insuring against livestock mortality: Predictive skill versus *RIB*

Livestock loss decreased with increasing *z*-scores up to about -1, for all indices (Fig 3). For *z*-scores higher than -1 there was hardly any effect on mortality. The two constrained linear regression models (lm0 and lm5) and the segmented regression model captured the same general relationship for mortality rates above 0.12. Differences below that rate, are irrelevant in this context, because there would not be a payout anyhow. The linear regression model fitted with all data shows a rather different response, as expected. The relationship between mortality and the indices is rather noisy, however, with high mortality events also occurring when forage supply conditions should be good (Fig 3), perhaps because of other causes such as disease.

The performance of the standard linear regression model was poor, as expected, both based on $R^2$ and for *RIB* (Fig 4). The median $R^2$ and *RIB* (across predictors) of lm0 was lower than lm5 and sm. The sm and lm5 models had the highest $R^2$ and *RIB* (Fig 4A and 4B). The three best predictors were MODIS NDVI (MD), log transformed MODIS NDVI (LMD), and untransformed rainfall (RN), with very little difference among them, although RN had more low quality outliers (Fig 4C and 4D).

The *RIB* and $R^2$ for the models were strongly related to each other, but there was noticeable variation in their ranking as shown in Fig 5 and S1 Tables 1, 2 in S1 Appendix. There were two models (sm and lm5)—each with three data regression combinations (RN, MD, and LMD)—with essentially the same *RIB* score of 0.50—that have an $R^2$ between 0.36 and 0.51. However, the model with the highest $R^2$ (0.51; lm5 with log transformed NOAA NDVI) had a lower *RIB* score (0.42). There was considerable variability in $R^2$, between 0.36 to 0.51, for the constrained piecewise regression models (lm0 and lm5) and segmented regression models that had a *RIB* of 0.42. Nonetheless, segmented regression with log transformed MODIS NDVI was clearly the best performing model when considering either *RIB* or R2.

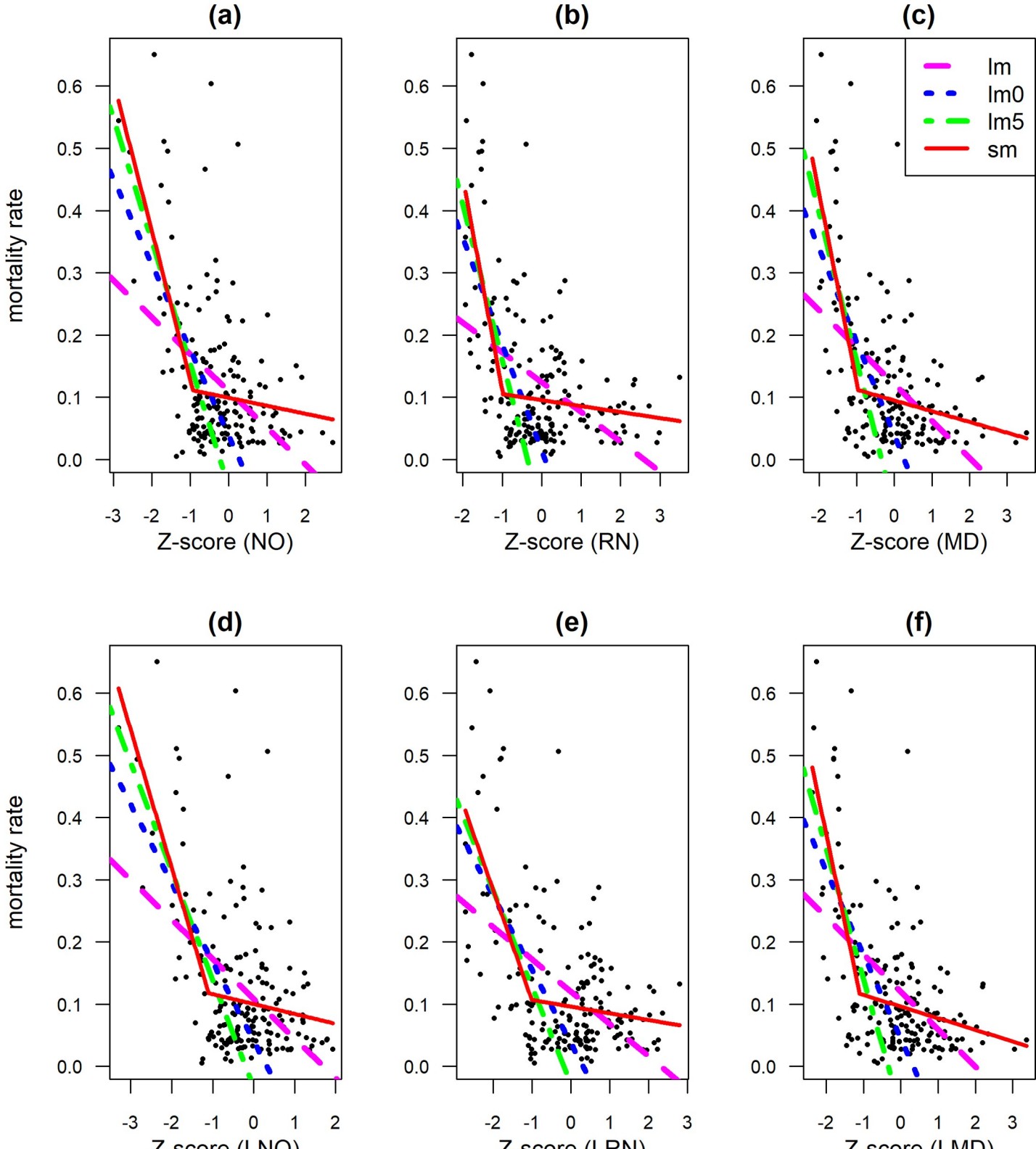

**Fig 3.** Livestock mortality in response to *z*-scores of (a) NOAA NDVI, (b) log(NOAA NDVI), (c) rainfall, (d) log(rainfall), (e) MODIS NDVI, and (f) log(MODIS NDVI) using linear regression (lm), piecewise linear regression when *z*-scores are less than zero (lm0) and -0.5 (lm5), and segmented regression (sm). Each observation corresponds to either the long or the short season of a sub-location in Marsabit.

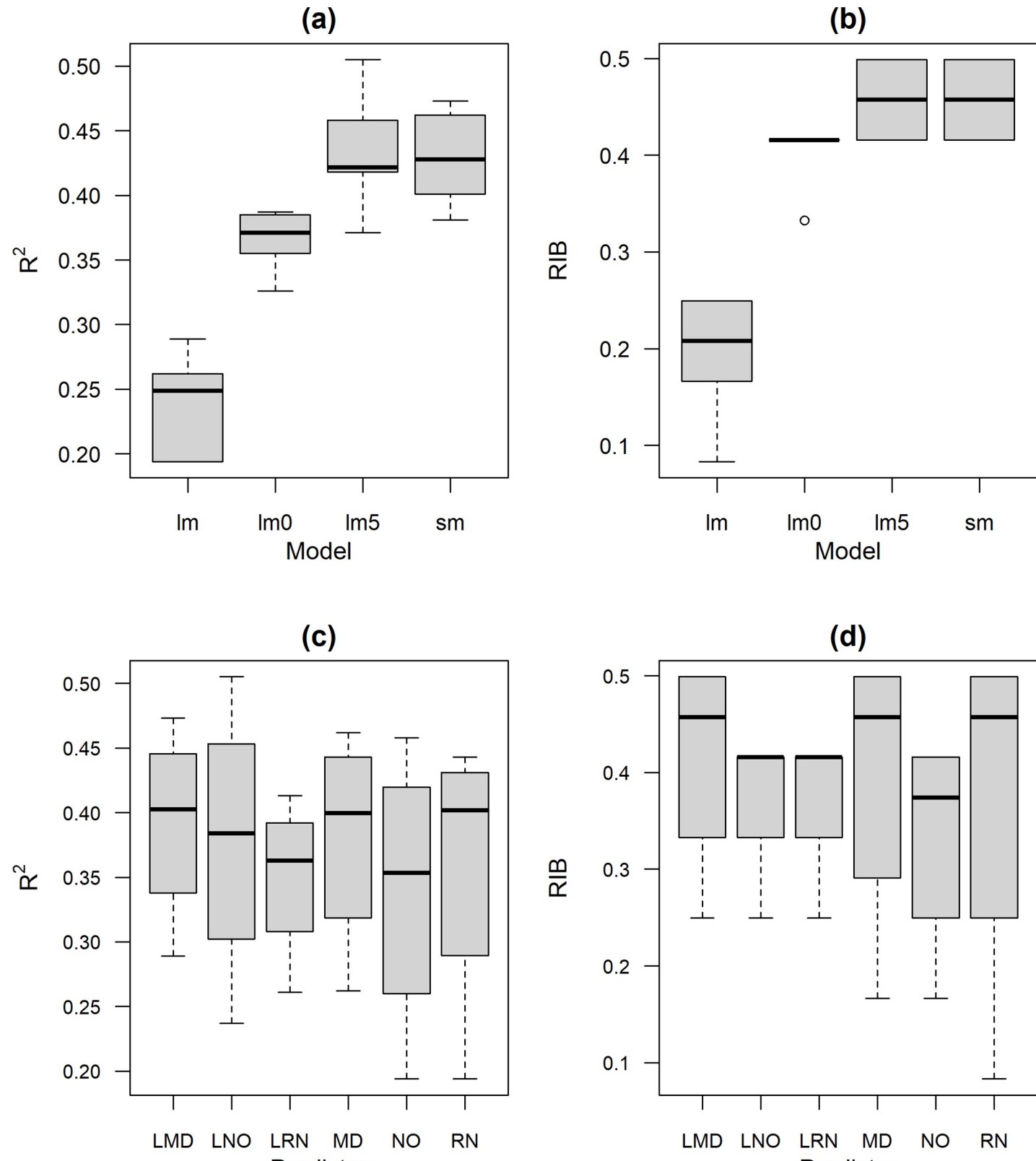

**Fig 4.** Assessment of the quality of an insurance index using $R^2$ and the Relative Insurance Benefit (*RIB*) measure for different regression models (a & b) and remote sensing predictor variables (c & d). Four regression models were used: linear (lm), piecewise linear with *z*-scores less than 0 (lm0), piecewise linear with *z*-scores less

than -0.5 (lm5), and segmented regression (sm). Data sources used as predictors were: Log MODIS NDVI (LMD), log NOAA NDVI (LNO), log rainfall (LRN), MODIS NDVI (MD), NOAA NDVI (NO), and rainfall (RN).

It is important to assure that the remote sensing models do not overfit the data, especially when using flexible model fitting algorithms. Using overfitted models would lead to overstating the quality of the index and the benefit of insurance. Cross-validation can be used to compare the difference between model fit to training and testing data, and to minimize this difference during model selection; or to compute more robust *RIB* values. As we used simple models with very few parameters, we are not especially concerned about it in our case, and there is very little scope for model simplification. A disadvantage of cross-validation is that it can create unstable estimates when working with small or modest sample sizes, as in our case. Nevertheless, we also evaluated our models using five-fold cross-validation. The goodness of fit statistics are lower (0.36 vs 0.28 on average), while the *RIB* values changed much less, on average (0.30 vs 0.27). Of the best models identified, log transformed MODIS-NDVI with segmented regression, had a slightly higher *RIB* and came out as the best model in cross-validation (see S2 Appendix for details).

As mentioned earlier while discussing livestock loss models, [36] suggest using conditional quantile analysis to characterize the insurance index function in preference to least squares methods. Applying their approach to our data (using the data that performed best, i.e., LMD), we found that the *RIB* for the conditional quantile estimate exceeds that of the least squares model by 5%, but lags the *RIB* for the segmented regression method (and other spline methods) by almost 25%. This finding suggests that this latter family of measures better captures the sensitivity of high mortality rates to the underlying forage index.

## Unpacking *RIB* versus predictive skill

To illustrate the power and pitfalls of remote sensing-based index insurance, we begin by taking a more careful look at the best remote sensing index contract (based on the sm model with log transformed MODIS NDVI) (Fig 6). Immediately apparent is the degree to which even this best index contract falls well short of the perfect contract. Indeed, the *RIB* of this contract is 0.5 (Fig 5), indicating that it achieves only 50% of the potential insurance benefit that could be achieved with a zero failure or perfect insurance contract. Fig 6 classifies seasonal payouts for this contract. True negative cases predominate, as would be expected since the contract is designed to pay out infrequently. Severe false negatives are highly damaging to the *RIB*, especially when actual losses are severe, the shadow price of money ($\lambda$) is high and payouts are nil to minimal. The two severe false negative events, when actual mortality reduced livestock holdings to about $400 and $500, and yet no indemnity was paid (leading to a large shortfall from the floor that would be provided by a perfect contract), cut deeply into the *RIB*. For most of the other severe loss events (with actual livestock holdings reduced to $300 to $700, the index contract offers modest to strong protection. As can be seen, there are also a set of severe false negative events that occur when actual losses are just below the contract strike point. These false negatives diminish the *RIB* because they occur somewhat frequently, but the shadow value of money is not so high compared to the severe false negative events that take place when actual losses are larger.

Finally, Fig 6 also shows a set of false positives. While the index overpays in these circumstances, the shadow value of money is low in these circumstances as no real losses occurred, and these events further undercut the *RIB* value as these events raise the contract premium.

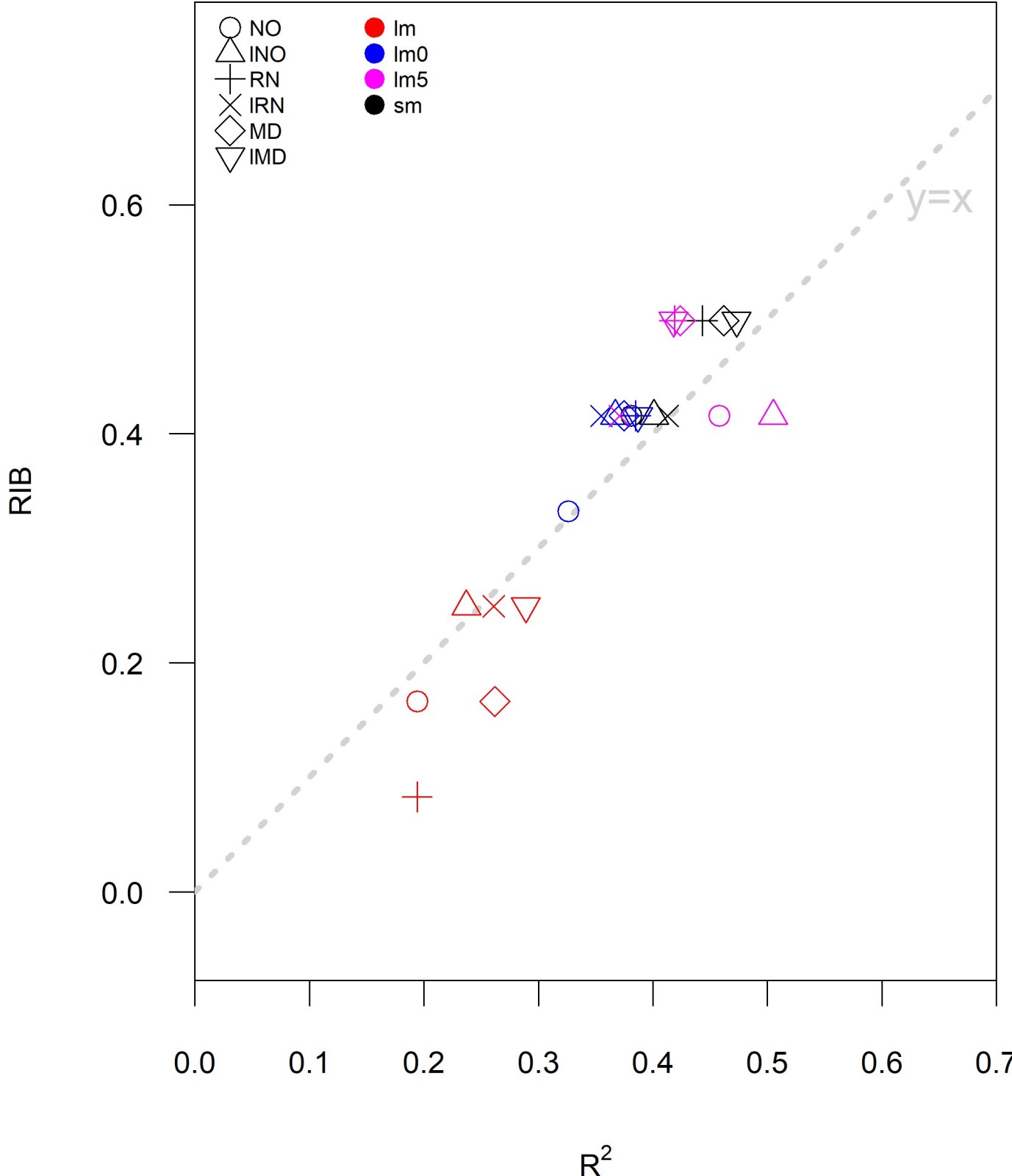

**Fig 5. Relationship between $R^2$ and the relative insurance benefit (*RIB*) for the 24 combination of four regression modelling approaches: Linear (lm), piecewise linear with z-scores less than 0 (lm0), piecewise linear with z-scores less than -0.5 (lm5), and segmented regression (sm), using z-scores derived from six data sources: Log MODIS NDVI (LMD), log NOAA NDVI (LNO), log rainfall (LRN), MODIS NDVI (MD), NOAA NDVI (NO), and rainfall (RN).**

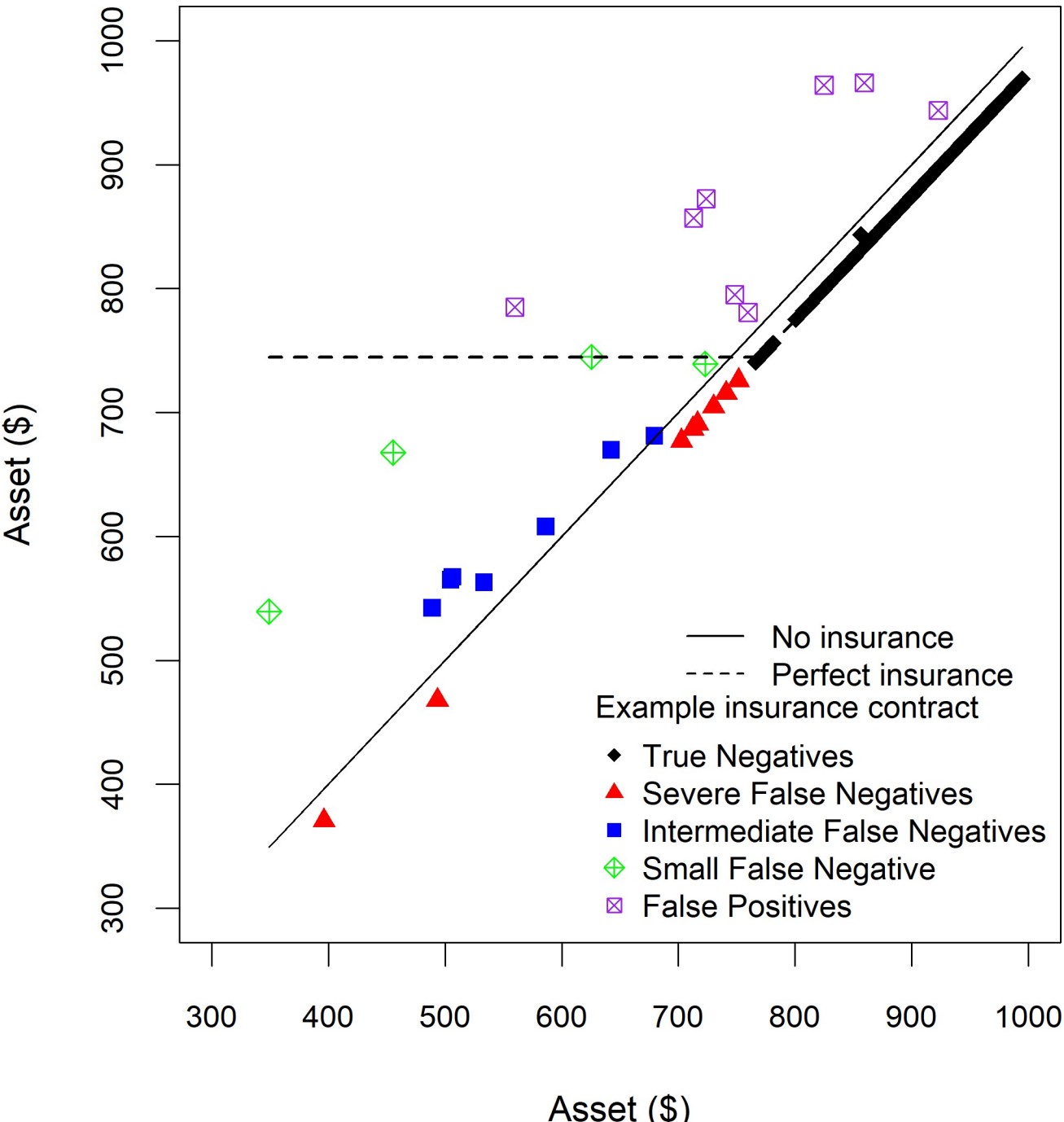

**Fig 6. Quality of a hypothetical index insurance contract based on the segmented (*sm*) regression model with LMD data (R$^2$ of 0.47 and *RIB* of 0.5) for 8 years and 15 sub-locations in Marsabit, Kenya (points), compared to a perfect insurance contract and to no insurance (lines).** The following classification was used for the contract: True Negatives: No payment due and none paid; Severe False Negatives: contract underpays by more than 30%; Intermediate False Negatives: contract underpays between 10% and 30%; Small False Negative: contract underpays less than 10%, and False Positives: contract overpays.

Every dollar received by the insured costs her $1.25, given our assumption of a 25% mark-up. When the shadow value of money is low (i.e., when a dollar is worth a dollar), it's well-being decreasing to pay $1.25 to get $1.

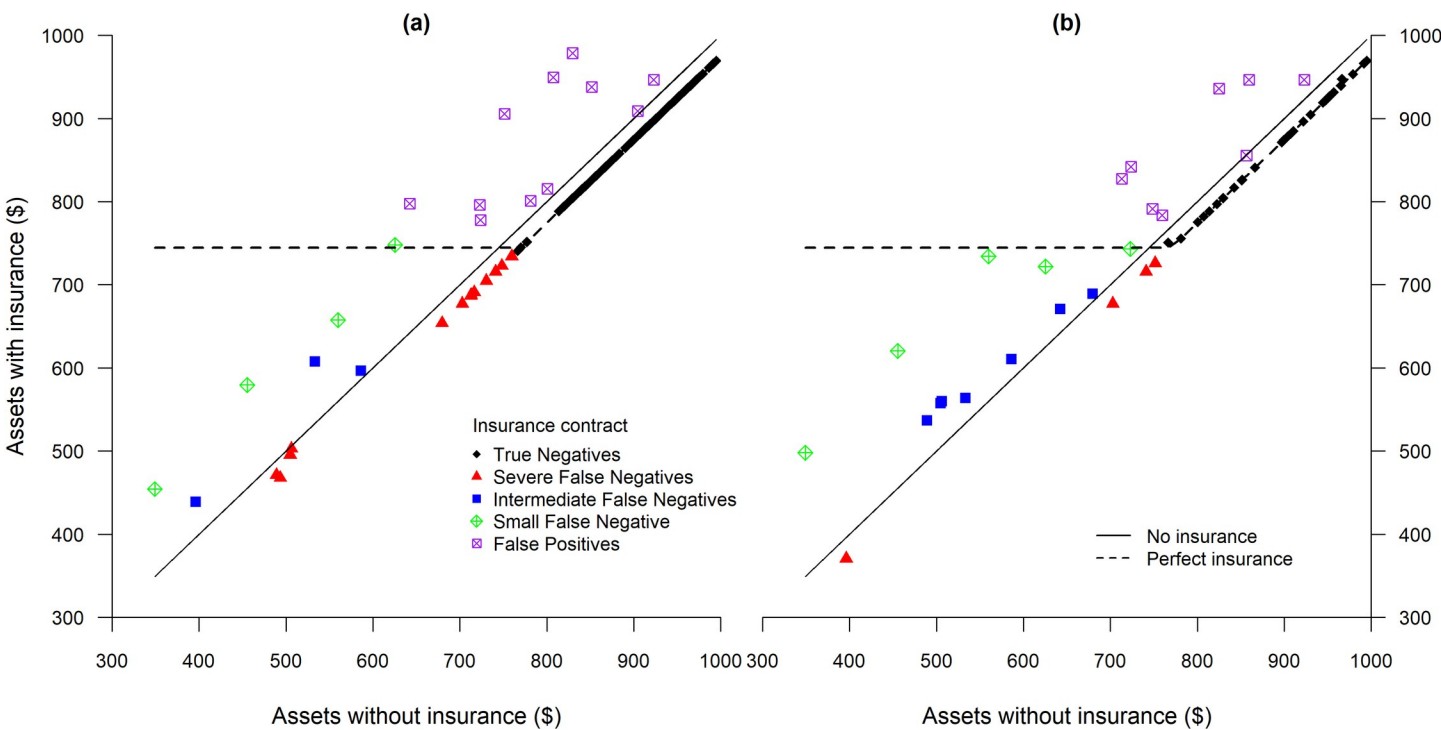

**Fig 7.** Comparison of perfect insurance contract versus two index insurance contracts with similar $R^2$ of 0.41 and *RIB* of 0.41 in (a) and 0.50 in (b). The index insurance contract in (a) in based on segmented (sm) model and lRN predictor while (b) uses piecewise linear with *z*-scores less than -0.5 (lm5) model and lMD predictor.

The two graphs in Fig 7 uses the same coding scheme to compare two contracts, based on sm model with LRN (sm-LRN) and lm5 model with LMD (lm5-LMD), that have almost identical predictive skill as judged by $R^2$ (Fig 8 illustrates the ability of the two contracts to correctly predict mortality rates), but very different *RIB* measures. The *RIB* for sm-LRN is 0.41, whereas the *RIB* for the contract based on lm5-LMD is 0.50, nearly identical to the contract illustrated in Fig 6.

Looking at the predictive skill revealed in the graphs in Fig 8, we can see that the contract based on LRN predictors and sm regression model (Fig 8A) is a more accurate predictor than the one designed using lm5 model with LMD predictors (Fig 8B) when losses are modest and below the strike point. However, the latter contract outperforms the former when mortality rates are high, as the former contracts tends to severely under-predict mortality under these circumstances. Hence, while the goodness of fit of both contacts is the same, the *RIB* measure asymmetrically penalizes the failures of the contract (Fig 8A) in those high mortality, bad states of the world. In other words, the prediction errors of Fig 8A contract in bad states of the world are not offset by its superior predictive performance in better states of the world.

Turning back to the graphs in Fig 7, we can see that there is only one instance (when shocks reduced livestock to about $400) when the sm-LRN contract (Fig 7A) barely outperforms that of lm5-lMD (Fig 7B). In all other severe loss events, the lm5-LMD contact outperforms the sm-LRN contract, increasing its comparative *RIB* measure. Finally, as can also be seen in the figure, the sm-LRN contract exhibits a few more false positive events, that again raise insurance costs without providing real insurance benefits, reducing the *RIB* of the contract.

## Discussion

We developed the *RIB*, a measure based on standard economic tools, to evaluate the quality of remote sensing-based indices for index insurance, and illustrated its use in a case study. The

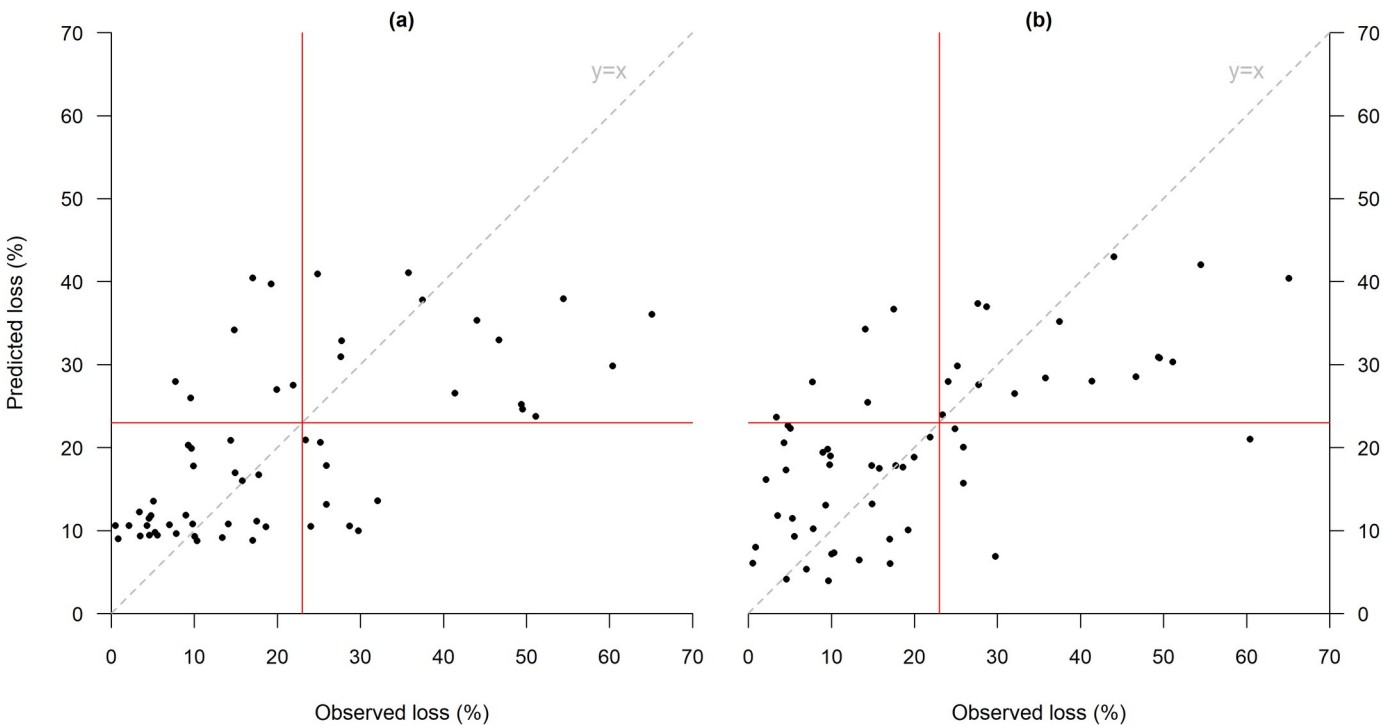

**Fig 8. Predictive skill of models used in Fig 7(A) and 7(B) respectively.** The red horizontal and vertical lines illustrates the mortality trigger at 23%.

*RIB* is a simple intuitive measure that expresses how much of the benefit of a perfect insurance contract is captured by an implementable, index insurance contract. In the case study, we showed that the model with the highest fit ($R^2$) is not necessarily the model with high insurance benefits. This is to be expected because better insurance welfare is predetermined by a model's ability to predict well the worst conditions that deserve most compensation. Consequently, it is important to note that if we only rely on conventional evaluation measures like $R^2$, we may not select the most suitable index.

The *RIB* essentially functions as an asymmetric loss function, penalizing estimation errors not only by their frequency $\pi$, but also according to $\lambda$, the shadow value of money (or neediness of households) when the estimation error occurs. The effect of shocks on $\lambda$ is non-linear, it increases sharply as conditions worsen for a household. The severe false negative events that occur when actual losses are high under the index insurance contracts analyzed in our case study hold the *RIB* measures for these contracts below 0.5.

The *RIB* is a direct measure of the quality of the index used. In our case-study we found *RIB* values of up to 0.5; indicating that the best of the index insurance contracts that we study and can achieve 50% of the benefit that could be generated by a perfect insurance contract. While the absolute benefit of an insurance program is of course the main concern when evaluating such programs, we could have tried to design a better program, for example by separating different types of livestock loss, and we could have tried to make it more realistic, for example by not giving equal weight to all sub-locations (some sub-locations will have more insured people than others). While this is relevant when evaluating a real program, it would not have helped us in illustrating *RIB*. The only important criterion was that the program designed had to have a positive outcome ($IB_r > 0$). If a program has no benefit with a perfect index, there is not much point in evaluating the actual index.

In other cases, it is useful to have a relative measure like the *RIB* for the remote sensing research supporting this work, as it is independent of other aspects of the design of insurance contracts. This makes *RIB* a simple measure that can be used to readily select remote sensing indices. It is also useful for more intricate analysis if, for example, different indices have different costs. For example, a remote sensing-based estimate of crop yield that depends on extensive field sampling (crop cuts) for model training may have a high quality in terms of $R^2$. But the *RIB* could also consider the higher (markup) cost of the resulting contract to evaluate whether the gain in model fit actually delivers a better contract. For these reasons, we believe that the *RIB* will allow to better compare studies and evaluate progress of remote sensing to support index insurance. To enable its use, the *RIB* computations have been implemented as an R package [37].

We evaluated a number of different predictor variables, including log transformed variables and models. Despite the log function transforming data distribution closer to normal distribution, it didn't not improve *RIB* and $R^2$ for rainfall. However, the log transformation improved the $R^2$ and *RIB* measures for the predictors with MODIS NDVI being the highest. While the comparison of different predictor variables and transformations was not the prime goal of this paper, the differences in model quality are notable, and highlight the need to compare models.

We assessed a hypothetical and stylized index insurance program. While none of the contracts we analyze have actually been implemented, they closely parallel the original index-based livestock insurance contract implemented in northern Kenya in 2009 [23]. In addition to the literature cited earlier, expected utility-based metrics have been used to evaluate the latest generation of IBLI contracts that prioritize methods that allow earlier detection and payment to producers who are suffering from rangeland stress [31].

These same methods can be applied to crop insurance contracts. Instead of estimating livestock mortality, crop yield can be estimated, and this is a very active research area for which considerable progress in obtaining reliable estimates has been reported [1, 38–40]. Such remote sensing based estimates of crop yield are commonly used in index insurance [1]. Crop insurance programs can also be evaluated with expected utility-based metrics. For example, [7] measured the quality of index-based insurance for rice producers in northern Tanzania. It would be important to further evaluate remote-sensing based indices for crop insurance with the *RIB* measure we propose because, similar to the case of livestock mortality, the quality of low-yield estimates are more important than the quality of high-yield estimates. One way to do this would be to use ground observed yields (e.g. estimated via crop cuts [41]) and to calibrate a remote sensing-based predictor ([39, 42]). Denoting predicted yields for any given farmer $i$ at a growing season $s$ in location $l$ as $\widehat{\mu}_{\ell s}$, we could write insurance payouts for a farmer with one unit of land as:

$$I_{ils}(k) = q \times \max(0, t \times \mu_l - \widehat{\mu}_{\ell s}) \tag{19}$$

where $t$ is again the trigger of deductible level, $\mu_\ell$ is the long-term average yield for location $\ell$ and $q$ is the unit value of the crop. Using this, or any other contract specification, we could then follow the procedures estimated above to estimate expected utility (or expected economic well-being) with and without insurance as the average of utilities computed from income using Eq (6). The insurance benefit for a farmer with a perfect contract (i.e., based on crop cut yield income estimates) and those estimated from a remotely sensed index (i.e., contract *J*) can now be estimated using Eq (11) and subsequently *RIB* using Eq (12).

In short, while our example contract has not been implemented, we used the livestock insurance case study to illustrate a method that can be applied to any contract for which it is

possible to backcast what payouts would have been in the period covered by the ground reference data. This includes contracts that adjust payments based on concerns about inter-temporal adverse selection (e.g., [13]) or other audit structures that insurers might use to moderate concerns about morally hazardous behaviour by the insured (e.g., see [43]). Irrespective of the measure used to evaluate the quality of an index it is necessary to have a time series of observations on what is insured (whether crop yield, livestock mortality, or something else) for good and especially for bad seasons. Observing extreme outcomes is especially important when using rainfall data, as the effect of low rainfall also depends on its distribution throughout the growing season. Such data may not exist and could take many years to collect. Although one can estimate the probabilities of the distribution of crop yield through simulation [1], long-term data is necessary to understand the value of index insurance and to evaluate ways to improve it. In the absence of such data, a reliable quantitative evaluation of the program may not be possible in which case the *RIB* cannot be estimated.

## Conclusions

We described an approach to evaluate the quality of an (remote sensing based) index for agricultural index insurance. The proposed measure, *RIB*, is derived from basic economic principles. While related to more conventional measures of predictive skill that might be used to evaluate the appropriateness of a remote sensing index for insurance purposes, the differences are often quite striking and are directly related to whether or not a particular index which actually make an insured better off in the worst of times. Across the 24 candidate insurance contracts we evaluated, none achieved more than 50% of the economic value that could be attained by a perfect insurance contract that exactly predicts average losses. This suggests there is ample room for improving existing remote sensing methods to make them work better as insurance and we hope that the *RIB* measure we propose will be of use for devising improved insurance indices.

## Supporting information

**S1 Appendix. Variability of risk preferences to insurance compensation.**
(PDF)

**S2 Appendix. Cross-validation results.**
(PDF)

**S1 Fig. Sensitivity of risk aversion to the value of an insurance contract evaluated over a range of 0–3.**
(TIF)

**S2 Fig.** Five-fold cross-validation based assessment of the quality of an insurance index using $R^2$ and the Relative Insurance Benefit (*RIB*) measure for different regression models (a & b) and remote sensing predictor variables (c & d). Four regression models were used: linear (lm), piecewise linear with *z*-scores less than 0 (lm0), piecewise linear with *z*-scores less than -0.5 (lm5), and segmented regression (sm). Data sources used as predictors were: Log MODIS NDVI (LMD), log NOAA NDVI (LNO), log rainfall (LRN), MODIS NDVI (MD), NOAA NDVI (NO), and rainfall (RN).
(TIF)

**S3 Fig. Internal vs five-fold cross-validated $R^2$ and *RIB* for the regression models used.**
(TIF)

## Acknowledgments

We thank Dr. Nathan Jensen (ILRI) for his guidance on IBLI survey data.

## Author Contributions

**Conceptualization:** Benson K. Kenduiywo, Michael R. Carter, Aniruddha Ghosh, Robert J. Hijmans.

**Data curation:** Benson K. Kenduiywo, Aniruddha Ghosh, Robert J. Hijmans.

**Methodology:** Benson K. Kenduiywo, Robert J. Hijmans.

**Software:** Benson K. Kenduiywo, Robert J. Hijmans.

**Supervision:** Robert J. Hijmans.

**Writing – original draft:** Benson K. Kenduiywo, Michael R. Carter, Robert J. Hijmans.

**Writing – review & editing:** Benson K. Kenduiywo, Michael R. Carter, Aniruddha Ghosh, Robert J. Hijmans.

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
