## [Decision Letter · Decision Letter 0]

24 May 2021

PONE-D-21-11491

Evaluating the Quality of Remote Sensing Products for Agricultural Index Insurance

PLOS ONE

Dear Dr. Kenduiywo,

Thank you for submitting your manuscript to PLOS ONE. After careful consideration, we feel that it has merit but does not fully meet PLOS ONE’s publication criteria as it currently stands. Therefore, we invite you to submit a revised version of the manuscript that addresses the points raised during the review process. Specifically

Metrics other than R2, RMSE: the paper seems to have ignored alternative metrics suggested by other authors in the literature. Please, review the suggestions by reviewers and either do a comparative analysis in terms of performance or point out how the suggested metric overcomes weaknesses of other metrics (apart from R2, RMSE) in the literatureMortality dataset: Please pay attention to comments on the description of the livestock mortality dataset and provide the necessary details either in the MS or as an appendix/supplementary materialAdditional or expanded analysis: a couple of suggestions for extending or conducting additional analysis have been made. Kindly consider these suggestions or provide reasons why they may not be necessary.Extension to crop insurance: Since the heading mentions "agricultural", i'll encourage some minimal discussion of how the recommended metric could be used or extended to crop insurance analysis as well

We look forward to receiving your revised manuscript.

Kind regards,

Gerald Forkuor

Academic Editor

PLOS ONE

Journal Requirements:

Reviewers' comments:

Reviewer's Responses to Questions

**Comments to the Author**

1. Is the manuscript technically sound, and do the data support the conclusions?

Reviewer #1: Partly

Reviewer #2: Yes

2. Has the statistical analysis been performed appropriately and rigorously? 

Reviewer #1: No

Reviewer #2: Yes

3. Have the authors made all data underlying the findings in their manuscript fully available?

Reviewer #1: Yes

Reviewer #2: Yes

4. Is the manuscript presented in an intelligible fashion and written in standard English?

Reviewer #1: Yes

Reviewer #2: Yes

5. Review Comments to the Author

Reviewer #1: Review: Evaluating the Quality of Remote Sensing Products for Agricultural Index Insurance, submitted to PlosOne.

The paper addresses a relevant question, i.e. how to improve the (assessment of) the quality of remote sensing based agricultural insurance. The paper is well-written, informative and technical solid. However, I have several concerns with the framing and analysis. For example, the embedding in the existing literature is not yet fully developed. Moreover, some expansions of the empirical analysis may be needed. I summarize key points below

The paper by Vroege et al 2021 gives a recent overview and reflection on the high potential of remote sensing for agricultural insurances.

The key claim and hook of this paper is that previous research in this field usually focused on measures like R2 or RMSE to evaluate index insurance schemes. This is wrong. Below just a quickly complied list with exceptions to your claim. For example: Heimfarth and Musshoff 2011 use a value at risk, Musshoff et al 2011 use a hedging effectiveness metric, Vedenov and Barnett, 2004 use semi-variances, Bucheli et al 2021 use expected utility and lower partial moments, Dalhaus et al 2020 use cumulative prospect theory. Along these lines, Conradt et al 2015 even suggest an econometric approach base don quantile regression to better account for low-yield events in designing index insurance schemes. You really need to incorporate previous literature in your paper. The arising question is: what is the value added of your manuscript, how does the suggested approach improve all that exists. I still think there is a potential value added, but this needs to be developed clearly.

In any case, you need to evaluate your proposed metric against more than R2. We already know the limitations of that. You may also use other metrics listed above for such comparison.

The developed metric implicitly assume a expected utility framework. We now that not all farmers have the same risk preferences, also aspects like loss aversion may differ substantially across farmers. Thus, is creating ‘one’ metric really realistic – shouldn’t this depend on preferences and be flexible to incorporate this? Again, you clearly need to motivate why this new approach is a value added.

Empirical analysis: i) if you focus especially on extreme events, quantile regression may be much more valuable (Conradt et al 2015), ii) you conduct your analysis in sample as far as I can see this from the paper. This has clear limitations, i.e. leads to biased results. Please provide an out if sample analysis.

Minor comments: i) the dataset is well known in the literature but still, using it deserves more details – maybe add an appendix, ii) estimation results shall be provided in tables, only figures are not sufficient, iii) how do you define ‘seasons’ you make your analysis for (page 8)? Be more specific in what you do. Again, maybe add an appendix with more details.

References:

Bucheli, J., Dalhaus, T., & Finger, R. (2021). The optimal drought index for designing weather index insurance. European Review of Agricultural Economics, in press https://doi.org/10.1093/erae/jbaa014

Conradt, S., Finger, R., Bokusheva, R. (2015). Tailored to the extremes: Quantile regression for index-based insurance contract design. Agricultural Economics 46: 1-11

Dalhaus T., Barnett B.J., Finger R. (2020) Behavioral weather insurance: Applying cumulative prospect theory to agricultural insurance design under narrow framing. PLos One 15(5): e0232267

Heimfarth, L. E., & Musshoff, O. (2011). Weather index‐based insurances for farmers in the north China Plain. Agricultural Finance Review.

Musshoff, O., Odening, M., & Xu, W. (2011). Management of climate risks in agriculture–will weather derivatives permeate?. Applied economics, 43(9), 1067-1077.

Vedenov, D. V., & Barnett, B. J. (2004). Efficiency of weather derivatives as primary crop insurance instruments. Journal of Agricultural and Resource Economics, 387-403.

Vroege, W., Vrieling, A., Finger, R. (2021). Satellite support to insure farmers against extreme droughts. Nature Food 2, 215–217

Reviewer #2: This paper develops an alternative to a standard R-squared fit to assess the functioning of index-based insurance product. To this end the authors create a new a metric (0<rib<1) assesses="" that="" the="">The paper is extremely well written, and fills a clear void in the literature and instruments available to researchers and the industry in assessing the functioning of index insurance products. This is a very welcome and timely contribution.

I have several comments, which I present below.

1. An alternative metric used in insurance is the so-called Catastrophic Performance Ratio (CPR), or the return to insurance relative to the premium paid, at different levels of the underlying index metric (e.g. different rainfall levels). The CPR is calculated by multiplying the probability of receiving a claim in case the farmer has catastrophic crop loss with the average amount of claim she receives in these cases and dividing this by the commercial premium. Given that to the best of my knowledge the CPR is the only alternative to date to assess the performance in negative states of an insurance, it would be important for this paper to acknowledge it and either 1) to some comparative analysis in terms of performance between CPR and RIB, or 2) at least describe how the RIB is different, and supposedly better, than the CPR.

2. The paper is very concise, which is appreciated, but there are two additions to their analysis that in my view can help the reader better understand the usefulness of the metric in question, and would really augment the contribution of the paper. The paper presents two NDVI measures and compares and contrasts their ability to generate functioning insurance as measured through the RIB. It also uses one rainfall dataset to the same end. It finds that the RIB helps distinguish ‘better’ indexes, and shows clearly that it does so better than a simple R-squared, as it puts more weight to severe false negatives, for instance. It finds that the segmented regression log transformed MODIS NDVI performs better than the other hypothetical indexes. This is quite illustrative of the power of RIB as it shows for instance that between MODIS NDVI and NOAA NDVI the preferred dataset to use is the former, using segmented regression modelling. It would be great to see a similar parallel analysis for rainfall, in the sense that for a similar proxy of weather we clearly see that certain models and certain datasets work better than others. To this end, I am wondering if it would be possible to complement the analysis thus far with an additional rainfall dataset: namely the ARC2 dataset that is used by other rainfall-based insurance products in Kenya.

3. Likewise, I would like to see at least one application of the RIB to an actual index. It would be great to have some insights into how the 24 (simplified) hypothetical models designed work in relationship to a real insurance. In fact, insurance payout triggers often conceal checks and balances that might be designed to avoid indemnifying policies that are the result of strategic behaviour etc. More generally speaking, actual insurance contracts might be more complex than the ones exposed in this paper. If the authors have access to them, it would be great to have at least one ‘true’ index applied as a ‘dry run’ on the same livestock mortality rates. Given that the context of the study is clearly pointing to IBLI, perhaps this real insurance could be a version of the IBLI index in Kenya, perhaps its’ very first iteration. Even more ideally, a first version of the IBLI could be compared to a more recent one (using the same mortality data)—to show how RIB can also pick up incremental improvements in the way real indexes are designed.

4. The livestock mortality dataset is described too hastily. How many surveys were collected in the 15 locations in total? How many data points, what is the average mortality rate in the dataset? More time should be spent describing the nature of this data.

5. The point above may also help to explain the extent to which the differences in RIB are to be expected if the same indexes are applied to the same locations but in 5 years time. In other words, to what extent can one say with certainty that a higher RIB means that an index alway functions better and to what extent may this be just by chance. More time should be spent discussing how the RIB can be used to discern index quality (some tests of significance?) and especially how it should NOT be used. This may require entering a discussion around power in the dataset you currently have. I am not asking you here to prove that these differences are systematic, I am asking mostly to discuss the issue openly and to caution against using the RIB to make assumptions that might be context and time specific.

6. If the data allows it (if the dataset is large enough) I would suggest randomly censoring half the data and calculating the RIB on the remaining random portion. Iterating this process several times and showing the RIB distribution can provide a useful impression of how data-dependent the RIB is, and how much we can expect it to vary by removing a few outliers.

7. Another important sensitivity analysis that should be performed and shown is that with respect to rho. It is acceptable that the authors pick a rho from literature and use it throughout their analysis, but they should show, at least in an appendix, what is the range of RIB values that we should expect when rho is varied within a reasonable range. Most importantly, they should discuss what is the likelihood that the relative RIB values of different indexes swap magnitude as the assumptions over rho vary.

8. It would be good to have a summary table somewhere describing the insurance performance in laymen terms. Which percentage of losses is covered by each index? Payout rates? True negatives? What is the percentage of false negatives and false positives? This will also help understand the improvement of RIB over R-squared more immediately. Ideally these rates can be presented both overall and at higher loss states.

9. The example with livestock mortality is compelling. However I am missing the complete picture that I was expecting from the title, referring to agricultural insurance. Authors should provide a discussion of how a similar index could be used for crop loss analysis. At the moment this is completely absent other than a brief mention in the discussion relative to the cost-benefit of crop cuts. I might be wrong, but the nature of livestock mortality makes it very suitable for a RIB-type index, while crop losses are much less lumpy and might be harder to assess—a prerequisite to make a functioning RIB. This should be discussed more, even in case the authors disagree, and the ‘complexities’ of constructing a RIB index should be weighted on its evident advantages. It is my hope that RIB will become a standard in the industry, well beyond livestock insurance, but for it to happen the necessary toolkit should be made clear across a variety of indexes, or at least the most common ones.

10. Equation 9, page 5, might need to be revised. Possibly the last superscript is N instead of P.</rib<1)>

6. PLOS authors have the option to publish the peer review history of their article (what does this mean?). If published, this will include your full peer review and any attached files.

Reviewer #1: No

Reviewer #2: **Yes: **francesco cecchi

---

## [Author Response · Author response to Decision Letter 0]

20 Jul 2021

Dear Editor,

We thank the reviewers for their efforts and suggestions to improve our manuscript. We have edited our manuscript to address the reviewers’ comments. Below you find the reviewers' comments (indented, and in black font) and our responses in blue font. So far we have made changes to the abstract and several sections in the document with additional supporting information in a bid to respond to reviewer comments. We hope that the information we provide will help improve our findings based on the valuable reviewer’s inputs.

Sincerely, 

Benson Kenduiywo

REVIEWER 1

The paper addresses a relevant question, i.e. how to improve the (assessment of) the quality of remote sensing based agricultural insurance. The paper is well-written, informative and technical solid. However, I have several concerns with the framing and analysis. For example, the embedding in the existing literature is not yet fully developed. Moreover, some expansions of the empirical analysis may be needed. I summarize key points below

 The paper by Vroege et al 2021 gives a recent overview and reflection on the high potential of remote sensing for agricultural insurances.

Thank you for this resource which we have now included in our Introduction.

 The key claim and hook of this paper is that previous research in this field usually focused on measures like R2 or RMSE to evaluate index insurance schemes. This is wrong. Below just a quickly complied list with exceptions to your claim. For example: Heimfarth and Musshoff 2011 use a value at risk, Musshoff et al 2011 use a hedging effectiveness metric, Vedenov and Barnett, 2004 use semi-variances, Bucheli et al 2021 use expected utility and lower partial moments, Dalhaus et al 2020 use cumulative prospect theory. Along these lines, Conradt et al 2015 even suggest an econometric approach base don quantile regression to better account for low-yield events in designing index insurance schemes. You really need to incorporate previous literature in your paper. The arising question is: what is the value added of your manuscript, how does the suggested approach improve all that exists. I still think there is a potential value added, but this needs to be developed clearly.

Our point of departure is that (1) remote sensing products to support index insurances are generally evaluated with measures like R2 or RMSE and that (2) while these measures are useful, we propose a more informative measure (the RIB). 

We were not sufficiently clear that the focus of our contribution is the evaluation of remote sensing products for index insurance. Our goal is not to argue for a particular economic framework for an overall evaluation of an index insurance program. 

The references provided by the reviewer are very useful as they illustrate different approaches for evaluating index insurance, and we now use these references in the manuscript. However, these references do not provide alternatives for the evaluation of remote sensing data used for index insurance. This is an important distinction, as our paper proposes a better way to evaluate remote sensing products. Rather than using a general measure (e.g. R2) we propose the RIB that allows the evaluation of the product's fitness for the specific purpose it is developed for. 

We now clarify that the RIB is independent of the overall (economic) framework to evaluate the benefits of an insurance program. In principle it can be applied to any framework to distinguish between the value of an insurance with a "perfect index" and the "actual index".

We believe that we use a solid economic framework to assess the quality of an index insurance program; but it is not our goal in this paper to argue that it is better than others. We use our approach to explain and illustrate the RIB --- to allow a deeper understanding of why RIB and R2 are different, beyond merely showing that they are different. We now point out in the Introduction and the Discussion that there are other approaches that have been used (citing the literature provided by the reviewer). It was an oversight to not do that in the earlier version and we thank the reviewer for pointing that out.

We have edited the Evaluation framework and Discussion sections to be much more clear and precise about our goals. Specifically, we state in the Evaluation framework section: lines 185-200,

A number of studies in the economics literature have employed different metrics which speak to the quality of index insurance contracts. These can be grouped into (i) Measures that examine the impact of the insurance on some feature of the probability distribution for wealth; and, (ii) Measures that are based on an explicit normative or welfare metric designed to capture the economic well-being of the insured household. Studies in the first category include [9], who study the hedging effectiveness of insurance, and a number of studies that look at the risk-reducing potential of insurance [10]–[13]. In a similar spirit, [14] study the catastrophic performance ratio (defined as expected payouts, normalized by the sum insured, in catastrophic, left tail states of the world). Somewhat similarly, [15] study the impact of insurance on lower partial moments of the probability distribution. While these approaches all offer valuable insights, they focus on changes in the left tail portion of the probability distribution. While what happens in the left tail is very important for the value of insurance, index contracts that incorrectly issue payouts in the right tail (false positives) are also damaging to the welfare value of insurance. This because as we discuss more below, it always costs more than $1 to get $1 of a payout. Paying more than a dollar to get a dollar makes sense if a dollar is worth more than a dollar, as it is in bad states of the world. In good states of the world, however, a dollar is worth only a dollar and paying, say, $1.25 to get a $1 does not improve economic welfare.

Lines 251-268

While expected utility theory has deep roots within the discipline of economics, its descriptive accuracy has been called into question by a range of experimental studies that show that some people do not make choices that conform with those that would maximize their expected utility. In efforts to accommodate systematic behavioral deviations from the predictions of expected utility theory [17] and [18] assembled alternative theories of how individuals make decisions in the face of risk, known as cumulative prospect theory (CPT) and rank dependent utility (RDU). These two alternative frameworks share the perspective that people make decisions using a probability weights that may differ systematically from objective probabilities. In addition, CPT assumes that individuals value losses differently than gains, creating a kink in the smooth utility function shown in Eq ([Disp-formula pone.0258215.e009]), where the kink occurs at what they call the reference point that distinguishes losses from gains. While such a reference point can be easily established in a behavioral experiment, its definition is less obvious in real world circumstances, making it more difficult to use CPT as a general tool for evaluating the quality of index insurance [36]. While CPT and RDU approaches have both been used to evaluate the value of index insurance to an individual (see [19], [20]), it is not obvious that a welfare metric based on misperception of probabilities is appropriate to judge and design insurance quality. Put differently, while these alternative approaches may be descriptively more accurate than expected utility theory in predicting insurance uptake, it is not apparent that they form the better basis for normative judgements [17] [36] especially if the source of objectively wrong probability weights is simple misperception.

and later in the discussion:

 We assessed a hypothetical and stylized index insurance program. While none of the contracts we analyze have actually been implemented, they closely parallel the original index-based livestock insurance contract implemented in northern Kenya in 2009 [22]. While remote sensing-based prediction of crop yields is more difficult than predicting rangeland forage, these same methods can be applied to crop insurance contracts. For example, [7] apply a related expected utility-based metric to measure the quality of index-based insurance for rice producers in northern Tanzania. In a similar vein, [30] show that the expected utility-based metrics we use can be utilized to evaluate the latest generation of IBLI contracts that prioritize methods that allow earlier detection and payment to producers who are suffering from rangeland stress. In short, while our example contracts have not been implemented, we use to illustrate a method that can be applied to any contract for which it is possible to backcast what payouts would have been in the period covered by the ground reference data. This includes contracts that adjust payments based on concerns about inter-temporal adverse selection (e.g., [12]) or other audit structures that insurers might use to moderate concerns about morally hazardous behaviour by the insured (e.g., see [38]).

 In any case, you need to evaluate your proposed metric against more than R2. We already know the limitations of that. You may also use other metrics listed above for such comparison.

We agree that R2 is not a good measure to evaluate insurance programs. However, it (and related measures like RMSE) is the dominant measure that is used to evaluate the quality of the remote sensing index. We are not evaluating the merits of different economic frameworks to evaluate index insurance. We were not clear enough about this narrow goal in the previous version, and we hope that our edits in the Introduction and Discussion now make this clear (see under point 2, above).

 The developed metric implicitly assume a expected utility framework. We know that not all farmers have the same risk preferences, also aspects like loss aversion may differ substantially across farmers. Thus, is creating ‘one’ metric really realistic – shouldn’t this depend on preferences and be flexible to incorporate this? Again, you clearly need to motivate why this new approach is a value added.

It is true that the economic framework we use to illustrate the RIB is based on Utility theory. We do not wish to imply that this is the only way to evaluate index insurance. We also mention that it can be extended by including variability in risk preferences as now depicted by S1 Fig in supplementary information section. We also more explicitly point out that the RIB can be applied to any overall index insurance evaluation metric (see our answer to Point 2).

 Empirical analysis: i) if you focus especially on extreme events, quantile regression may be much more valuable (Conradt et al 2015), 

We like quantile regression, and we tried it with our data (as noted in footnotes 5 and 6 in the interest of space), and it performed very poorly. That is because in this application, we want an index that predicts the mean effect of low NDVI on mortality, not one that models the outliers (a high quantile). When you consider Figure 4, this is not so surprising; the standard models fit the data about as good as can be expected given the (noisy) data. Moving the regression line up to say the 0.9 quantile, leads to a lot of false positives.

 ii) you conduct your analysis in sample as far as I can see this from the paper. This has clear limitations, i.e. leads to biased results. Please provide an out if sample analysis.

Out-of-sample model evaluation (cross-validation) is important when using flexible modeling algorithms that may overfit the data. In our case we use single-variable regression models. This is not a flexible modeling approach at all, and overfitting is not a concern (James, Witten, Hastie & Tibshirani, 2013. An Introduction to Statistical Learning. Springer).

 Minor comments: i) the dataset is well known in the literature but still, using it deserves more details – maybe add an appendix, 

Our goal is methodological, not to go into the details of the example data set, but we have added more information in the Methods section that now reads:

We used livestock mortality data collected from households in 15 sub-locations of Marsabit by the International Livestock Research Institute [26]. Annual household surveys in Marsabit were conducted between October and November to cover the years 2008–2015. Each of the annual surveys was carried out after IBLI insurance sales in each corresponding season (i.e. LRLD or SRSD). The main purpose of the data collection was to help develop and monitor the Index Based Livestock Insurance (IBLI) program in Marsabit [26]. The number of household surveyed per year varied between 1739 and 3549. Surveys were carried out between October and November and included questions related to household demographic information, livestock production, livelihoods activities and income sources, expenditures and consumption, health and educational outcomes, assets, access to credit, market interaction, and community poverty rates. These data have been described in detail by [26] and also used by [5], [11], [12], [22], [27]–[30] to design and evaluate impact of IBLI program.

 Of prime interest to our study was the livestock loss data. Livestock loss can occur for several reasons, but loss from drought-related starvation was the primary reason, followed, at some distance by disease. To combine data on different livestock species, the livestock numbers were transformed to the equivalent TLUs. We averaged mortality data by a given sub-location and season. Overall the average morality in rate from the survey data was 13%.

 ii) estimation results shall be provided in tables, only figures are not sufficient, 

We follow the convention (and policy in most journals, but for as far as we can not of PLoS-ONE) to show a set of results in either a figure or in a table; but not in both. Nevertheless, we have now added the main results in a table as well, in the supplementary information section. 

 iii) how do you define ‘seasons’ you make your analysis for (page 8)? Be more specific in what you do. Again, maybe add an appendix with more details.

The seasons are described in the Methods section (under study area, [line 342-354].

References:

Bucheli, J., Dalhaus, T., & Finger, R. (2021). The optimal drought index for designing weather index insurance. European Review of Agricultural Economics, in press https://doi.org/10.1093/erae/jbaa014

Conradt, S., Finger, R., Bokusheva, R. (2015). Tailored to the extremes: Quantile regression for index-based insurance contract design. Agricultural Economics 46: 1-11

Dalhaus T., Barnett B.J., Finger R. (2020) Behavioral weather insurance: Applying cumulative prospect theory to agricultural insurance design under narrow framing. PLos One 15(5): e0232267 

Heimfarth, L. E., & Musshoff, O. (2011). Weather index‐based insurances for farmers in the north China Plain. Agricultural Finance Review.

Musshoff, O., Odening, M., & Xu, W. (2011). Management of climate risks in agriculture–will weather derivatives permeate?. Applied economics, 43(9), 1067-1077.

Vedenov, D. V., & Barnett, B. J. (2004). Efficiency of weather derivatives as primary crop insurance instruments. Journal of Agricultural and Resource Economics, 387-403.

Vroege, W., Vrieling, A., Finger, R. (2021). Satellite support to insure farmers against extreme droughts. Nature Food 2, 215–217

REVIEWER 2

This paper develops an alternative to a standard R-squared fit to assess the functioning of index-based insurance product. To this end the authors create a new a metric (0The paper is extremely well written, and fills a clear void in the literature and instruments available to researchers and the industry in assessing the functioning of index insurance products. This is a very welcome and timely contribution.

I have several comments, which I present below.

1. An alternative metric used in insurance is the so-called Catastrophic Performance Ratio (CPR), or the return to insurance relative to the premium paid, at different levels of the underlying index metric (e.g. different rainfall levels). The CPR is calculated by multiplying the probability of receiving a claim in case the farmer has catastrophic crop loss with the average amount of claim she receives in these cases and dividing this by the commercial premium. Given that to the best of my knowledge the CPR is the only alternative to date to assess the performance in negative states of an insurance, it would be important for this paper to acknowledge it and either 1) to some comparative analysis in terms of performance between CPR and RIB, or 2) at least describe how the RIB is different, and supposedly better, than the CPR.

The RIB evaluates the performance of the remote sensing product within the context of an index insurance program evaluation metric. It expresses how much the insurance program deteriorates due to imperfections in the index. Therefore, the RIB could be applied to the CPR measure as well. In your example, RIB would be 1 if the (rainfall) index has no error. It thus allows to separate the quality of the design of a program (assuming a perfect index) from the quality of the index itself. You could have a well designed program but a worthless index. Or (at least theoretically) a very good index, but a program that still has little benefit to consumers. 

We now make this much more explicit in the text (lines 185-200) (also see response to reviewer 1) and we refer to the CPR as an alternative approach that could be used with the RIB.

2. The paper is very concise, which is appreciated, but there are two additions to their analysis that in my view can help the reader better understand the usefulness of the metric in question, and would really augment the contribution of the paper. The paper presents two NDVI measures and compares and contrasts their ability to generate functioning insurance as measured through the RIB. It also uses one rainfall dataset to the same end. It finds that the RIB helps distinguish ‘better’ indexes, and shows clearly that it does so better than a simple R-squared, as it puts more weight to severe false negatives, for instance. It finds that the segmented regression log transformed MODIS NDVI performs better than the other hypothetical indexes. This is quite illustrative of the power of RIB as it shows for instance that between MODIS NDVI and NOAA NDVI the preferred dataset to use is the former, using segmented regression modelling. It would be great to see a similar parallel analysis for rainfall, in the sense that for a similar proxy of weather we clearly see that certain models and certain datasets work better than others. To this end, I am wondering if it would be possible to complement the analysis thus far with an additional rainfall dataset: namely the ARC2 dataset that is used by other rainfall-based insurance products in Kenya.

The goal of the paper is to propose the RIB metric as a way to evaluate the quality of a remote sensing index within the framework of economic evaluation. We could have simulated some data to show this, but we thought that it would be better to use real and commonly used data. There are indeed alternative rainfall datasets that one can use. Our experience suggests that while there is variation between them, they tend to perform similarly --- because they all pick up the dry and the wet years; the problem is more one of relating rainfall to productivity because of the variability in rainfall amounts. But even so, our goal is not to strictly compare these different data sources to create an index; we just want to be able to show that there is variability.

3. Likewise, I would like to see at least one application of the RIB to an actual index. It would be great to have some insights into how the 24 (simplified) hypothetical models designed work in relationship to a real insurance. In fact, insurance payout triggers often conceal checks and balances that might be designed to avoid indemnifying policies that are the result of strategic behaviour etc. More generally speaking, actual insurance contracts might be more complex than the ones exposed in this paper. If the authors have access to them, it would be great to have at least one ‘true’ index applied as a ‘dry run’ on the same livestock mortality rates. Given that the context of the study is clearly pointing to IBLI, perhaps this real insurance could be a version of the IBLI index in Kenya, perhaps its’ very first iteration. Even more ideally, a first version of the IBLI could be compared to a more recent one (using the same mortality data)—to show how RIB can also pick up incremental improvements in the way real indexes are designed.

That is an interesting suggestion, but it is far from trivial and we would require a separate paper (and collaboration with insurers) to address it. Again, our goal is to present a new statistic that can be used to evaluate the quality of a remote sensing product used as the index in an index insurance scheme. While we use real (instead of simulated) data, our goal is not to evaluate an insurance program (nor to make strong statements about the best index, as argued above). The point that real programs are more complex is well taken; but that should not affect the ability to compute the RIB. We have clarified this point in the manuscript using the following text:

In “Livestock loss models” section we added: 

Note that the procedures we are using parallel the actual approach that was used to create the original IBLI contract in northern Kenya, although that original exercise was based on different data sources and a more restricted set of remote sensing based measures (see Chantarat et al., 2013). 

This segmented model is very similar to the model used for the initial IBLI contract in northern Kenya (Chantarat et al., 2013). While that earlier work was not guided by the sort of explicit quality standard we propose here, we shall see that this model is in fact our best performing model using the RIB standard.

And in the “Discussion” section we also added:

 We assessed a hypothetical and stylized index insurance program. While none of the contracts we analyze have actually been implemented, they closely parallel the original index-based livestock insurance contract implemented in northern Kenya in 2009 [22]. While remote sensing-based prediction of crop yields is more difficult than predicting rangeland forage, these same methods can be applied to crop insurance contracts. For example, [7] apply a related expected utility-based metric to measure the quality of index-based insurance for rice producers in northern Tanzania. In a similar vein, [30] show that the expected utility-based metrics we use can be utilized to evaluate the latest generation of IBLI contracts that prioritize methods that allow earlier detection and payment to producers who are suffering from rangeland stress. In short, while our example contracts have not been implemented, we use to illustrate a method that can be applied to any contract for which it is possible to backcast what payouts would have been in the period covered by the ground reference data. This includes contracts that adjust payments based on concerns about inter-temporal adverse selection (e.g., [12]) or other audit structures that insurers might use to moderate concerns about morally hazardous behaviour by the insured (e.g., see [38]).

4. The livestock mortality dataset is described too hastily. How many surveys were collected in the 15 locations in total? How many data points, what is the average mortality rate in the dataset? More time should be spent describing the nature of this data.

We have expanded the description of the data set [See response to Reviewer 1, #7 ] 

5. The point above may also help to explain the extent to which the differences in RIB are to be expected if the same indexes are applied to the same locations but in 5 years time. In other words, to what extent can one say with certainty that a higher RIB means that an index alway functions better and to what extent may this be just by chance. More time should be spent discussing how the RIB can be used to discern index quality (some tests of significance?) and especially how it should NOT be used. This may require entering a discussion around power in the dataset you currently have. I am not asking you here to prove that these differences are systematic, I am asking mostly to discuss the issue openly and to caution against using the RIB to make assumptions that might be context and time specific.

Our ability to evaluate an insurance program (and hence also our ability to compute a meaningful RIB) depends on the data available. While the RIB is conditional on the overall measure, it still is relevant to address this issue. At the end of the Discussion we state that:

Irrespective of the measure used to evaluate the quality of an index it is necessary to have a time series of observations on what is insured (whether crop yield, livestock mortality, or something else) for good and especially for bad seasons. Observing extreme outcomes is especially important when using rainfall data, as the effect of low rainfall also depends on its distribution throughout the growing season. Such data may not exist and could take many years to collect. Although one can estimate the probabilities of the distribution of crop yield through simulation [1], long-term data is necessary to understand the value of index insurance and to evaluate ways to improve it. In the absence of such data, a reliable quantitative evaluation of the program may not be possible in which case the RIB cannot be estimated.

6. If the data allows it (if the dataset is large enough) I would suggest randomly censoring half the data and calculating the RIB on the remaining random portion. Iterating this process several times and showing the RIB distribution can provide a useful impression of how data-dependent the RIB is, and how much we can expect it to vary by removing a few outliers.

We discuss the general idea (see the point above), but we do not implement it here because (a) our dataset is not very large and (b) this can have a large effect on the ability to evaluate an insurance program; this indirectly affects the RIB, but our main goal is to show how to evaluate a remote sensing index given an insurance evaluation protocol. 

7. Another important sensitivity analysis that should be performed and shown is that with respect to rho. It is acceptable that the authors pick a rho from literature and use it throughout their analysis, but they should show, at least in an appendix, what is the range of RIB values that we should expect when rho is varied within a reasonable range. Most importantly, they should discuss what is the likelihood that the relative RIB values of different indexes swap magnitude as the assumptions over rho vary.

We have added figure (S1 Fig) in the supporting information section to illustrate the variation of rho. The figure is explained as follows:

 S1 Fig shows the impact of risk aversion on the certainty equivalent components that combine to create the RIB measure in Eq (Error! Reference source not found.). The dashed (red) line displays CE^N and the dot-dashed (green) line displays CE^P, the certainty equivalent for perfect insurance. Because the insurance is marked up over the actuarially fair price, a risk neutral agent (ρ=0) would be better off without insurance. Perfect insurance offers positive benefits for all risk aversion levels beyond about 0.4. The index insurance contract generates certainty equivalent levels shown by the solid (blue) curve. It offers no insurance benefit at risk aversion levels below 0.6. At the moderate risk aversion level of 1.5 used in this paper’s RIB calculations, we see that the index provides roughly half the gain of the perfect contract as the solid line lies mid-way between CE^P and CE^N . At every higher level of risk aversion, the RIB for the index contract declines slightly, reflecting the fact that even the best index contract occasionally fails to correctly compensate severe losses. 

8. It would be good to have a summary table somewhere describing the insurance performance in laymen terms. Which percentage of losses is covered by each index? Payout rates? True negatives? What is the percentage of false negatives and false positives? This will also help understand the improvement of RIB over R-squared more immediately. Ideally these rates can be presented both overall and at higher loss states.

We follow the convention (and policy in most journals, but for as far as we can not of PLoS-ONE) to show a set of results in either a figure or in a table; but not in both. Nevertheless, we have now added the main results in a table as well, in the supplementary information section and moreover Figures 4, 6 and 7 captures well the requested information. 

9. The example with livestock mortality is compelling. However I am missing the complete picture that I was expecting from the title, referring to agricultural insurance. Authors should provide a discussion of how a similar index could be used for crop loss analysis. At the moment this is completely absent other than a brief mention in the discussion relative to the cost-benefit of crop cuts. I might be wrong, but the nature of livestock mortality makes it very suitable for a RIB-type index, while crop losses are much less lumpy and might be harder to assess—a prerequisite to make a functioning RIB. This should be discussed more, even in case the authors disagree, and the ‘complexities’ of constructing a RIB index should be weighted on its evident advantages. It is my hope that RIB will become a standard in the industry, well beyond livestock insurance, but for it to happen the necessary toolkit should be made clear across a variety of indexes, or at least the most common ones.

There is no mathematical reason why computing an RIB would be challenging for crops. It may very well be that crop losses are more difficult to assess than livestock losses. Whether (or rather, where, or for what crop) this is true or not is hard to say; there is no literature that we know of that makes this comparison. We agree that it can be difficult to make a reliable index for crop loss. But that is, in fact, what inspires our work. We believe that many proposed and actual index insurance programs are indeed not viable. This can be hard to prove or disprove if available data is limited. What we propose is a way to evaluate such programs if data are available --- or can partly be simulated. We now explicitly state that "We used a case study on livestock insurance, but the economic framework used, and the RIB also apply to crop insurance"

10. Equation 9, page 5, might need to be revised. Possibly the last superscript is N instead of P.

Thank you, we have corrected the Equation as indicated below:

 IB^P≡EU^P-EU^N (1)

---

## [Decision Letter · Decision Letter 1]

2 Aug 2021

PONE-D-21-11491R1

Evaluating the Quality of Remote Sensing Products for Agricultural Index Insurance

PLOS ONE

Dear Dr. Kenduiywo,

Thank you for submitting your manuscript to PLOS ONE. After careful consideration, we feel that it has merit but does not fully meet PLOS ONE’s publication criteria as it currently stands. Therefore, we invite you to submit a revised version of the manuscript that addresses the points raised during the review process.

Please re-consider the request to provide an out-of-sample analysis as this is clearly known to provide more robust and trustworthy model assessment than in-sample. That aside, an out-of-sample analysis could also address the concern of how data dependent the RIB is (raised by reviewer 2), irrespective of the size of the data. While i appreciate the explanation that the modelling approach is not susceptible to over-fitting, i would encourage the use of out-of-sample to clear any doubts of biased results.I think that the response "We used a case study on livestock insurance, but the economic framework used, and the RIB also apply to crop insurance" is a bit simplistic and requires additional details regarding the applicability in crop insurance. Yes, it should be theoretically possible to apply it to crop insurance, but as the two systems are markedly different, e.g. in terms of assessing losses, authors may provide some few lines to guide the reader or potential users on extending this to crop insurance. Else, you may decide to revise the title of the paper to "livestock index insurance" instead of "agricultural index insurance".

We look forward to receiving your revised manuscript.

Kind regards,

Gerald Forkuor

Academic Editor

PLOS ONE

Journal Requirements:

Reviewers' comments:

Reviewer's Responses to Questions

**Comments to the Author**

1. If the authors have adequately addressed your comments raised in a previous round of review and you feel that this manuscript is now acceptable for publication, you may indicate that here to bypass the “Comments to the Author” section, enter your conflict of interest statement in the “Confidential to Editor” section, and submit your "Accept" recommendation.

Reviewer #1: (No Response)

2. Is the manuscript technically sound, and do the data support the conclusions?

Reviewer #1: Partly

3. Has the statistical analysis been performed appropriately and rigorously? 

Reviewer #1: Yes

4. Have the authors made all data underlying the findings in their manuscript fully available?

Reviewer #1: Yes

5. Is the manuscript presented in an intelligible fashion and written in standard English?

Reviewer #1: Yes

6. Review Comments to the Author

Reviewer #1: Big thanks to the authors for the thorough revision. That helped clarifying many things. I have two remaining issues, where I am not satisfied by the responses given.

First, the authors acknowledge that similar lines of thinking (using alternative metrics cp to R2) have been already used widely in the literature on index insurance. This is standard in many ag econ journals. BUT. the big gap is that it has not been done in the remote sensing related insurance applications. I do not see the difference here. Technically, it does not matter what index I used if I talk about assessment in economic terms. Please tone down the description of your own contribution with this paper, adjust language.

Second, I do not fully get your response on out of sample performance. It does not matter what estimator etc you use. Within sample assessment (as you do it) is biased in terms of assessing the economic performance of an insurance solution, i.e. it will be usually too optimistic if doing all in-sample. Thus, also comparisons and assessments presented here are likely biased. An out-of-sample procedure (again, widely used in ag econ journals) can address these concerns and add robustness to your conclusions. As you have sufficient data, it shall not be an issue I believe.

7. PLOS authors have the option to publish the peer review history of their article (what does this mean?). If published, this will include your full peer review and any attached files.

Reviewer #1: No

---

## [Author Response · Author response to Decision Letter 1]

14 Sep 2021

Dear Editor,

Thank you for your suggestions including those of the reviewers to improve our manuscript titled “Evaluating the Quality of Remote Sensing Products for Agricultural Index Insurance”. We have edited the manuscript and attached on the submission system (response to reviewers) you find the comments (indented in italics and black font) and our responses in blue font.

Sincerely, 

Benson Kenduiywo 

(on-behalf of all the co-authors)

---

## [Editor Report · Decision Letter 2]

22 Sep 2021

Evaluating the Quality of Remote Sensing Products for Agricultural Index Insurance <o:p></o:p>

PONE-D-21-11491R2

Dear Dr. Kenduiywo,

We’re pleased to inform you that your manuscript has been judged scientifically suitable for publication and will be formally accepted for publication once it meets all outstanding technical requirements.

Kind regards,

Gerald Forkuor

Academic Editor

PLOS ONE

Additional Editor Comments (optional):

Thank you for paying attention to the comments and performing extra analysis.
---

## [Editor Report · Acceptance letter]

30 Sep 2021

PONE-D-21-11491R2 

Evaluating the Quality of Remote Sensing Products for Agricultural Index Insurance 

Dear Dr. Kenduiywo:

I'm pleased to inform you that your manuscript has been deemed suitable for publication in PLOS ONE. Congratulations! Your manuscript is now with our production department. 

Kind regards, 

on behalf of

Dr. Gerald Forkuor 

Academic Editor

PLOS ONE